# A Comprehensive Analysis of Chemical Composition and Anti-Inflammatory Effects of Cassava Leaf Extracts in Two Varieties in *Manihot esculenta* Crantz

**DOI:** 10.3390/ijms26094140

**Published:** 2025-04-27

**Authors:** Jie Cai, Wenli Zhu, Jingjing Xue, Yanqing Ma, Kaimian Li, Lanyue Zhang, Oluwaseun Olayemi Aluko, Songbi Chen, Xiuqin Luo, Feifei An

**Affiliations:** 1Tropical Crops Genetic Resources Institute, Chinese Academy of Tropical Agricultural Science/Key Laboratory of Agriculture for Germplasm Resources Conservation and Utilization of Cassava, Haikou 571101, China; caijie@catas.cn (J.C.); zhuwenbamboo@catas.cn (W.Z.); xuetao608@catas.cn (J.X.); likaimian@sohu.com (K.L.); aluko.oluseun@gmail.com (O.O.A.); songbichen@catas.cn (S.C.); 2National Key Laboratory for Tropical Crop Breeding, Sanya 572025, China; 3Sanya Research Institute, Chinese Academy of Tropical Agricultural Sciences, Sanya 572025, China; 4Key Laboratory of Tropical Crops Germplasm Resources Genetic Improvement and Innovation of Hainan Province, Haikou 571101, China; 5School of Biomedical and Pharmaceutical Sciences, Guangdong Provincial Key Laboratory of Plant Resources Biorefinery, Guangdong University of Technology, Guangzhou 510006, China; 3222008434@mail2.gdut.edu.cn (Y.M.); zhanglanyue@gdut.edu.cn (L.Z.)

**Keywords:** cassava, leaf extracts, chemical composition, anti-inflammatory, transcriptomics analysis

## Abstract

Cassava is a tropical tuberous root crop, feeding over a billion people globally. However, research on the chemical composition and bioactive effects of cassava leaves remains scarce. Two specific varieties of South China No. 9 (green leaves (G.L.)) and South China No. 20 (purple leaves (P.L.)) were investigated in this study. The components of G.L. and P.L. were analyzed under different extraction methods using ultra-performance liquid chromatography time-of-flight mass spectrometry (UPLC-Q-TOF/MS). Results showed that cassava leaf extracts are rich in bioactive metabolites such as D-(+)-mannose, trigonelline, rutin, kaempferol-3-*O*-rutinoside, and oleamide. To assess the anti-inflammatory efficacy of bioactive compounds, animal models were established. Compared to the histamine group (NA), the group treated with the extracts had reduced epidermal thickness in hematoxylin and eosin (HE) staining. Further analysis revealed a drastic reduction in the number of mast cells in toluidine blue (TB) staining and expression levels of inflammatory cytokines (IL-17 and TNF-α) in immunohistochemistry (IHC) staining. The ethanolic extracts from the leaves demonstrated potent anti-inflammatory activities, with the extract from G.L. surpassing that from P.L. Transcriptomic analyses propose that the anti-inflammatory effects of cassava leaves may be related to the modulation of genes involved in mast cell activation, such as *Cma1*, *Cpa3*, and *Fn1*, among others. Network pharmacology unveiled that the extract of cassava leaves modulates pathways associated with apoptosis, inflammation, and metabolism. Molecular docking revealed strong binding interactions between 1-stearoylglycerol and oleamide from cassava leaves extracts and the proteins of AKT1, TNF, and BRAF. Overall, cassava leaf extracts seem to be a promising natural anti-inflammatory agent.

## 1. Introduction

Cassava (*Manihot esculenta*) is an important root crop in tropical and subtropical regions, feeding over a billion people in the world [1,2]. Cassava leaves exhibit phytotoxicity; the use of cassava leaves is limited by cyanide toxicity [3], which must be removed or reduced before they can be used as food or animal feed or consumed as vegetables [4,5]. Cassava leaves are rich in antioxidants (flavonoids, terpenoids) with antibacterial and anti-inflammatory properties, making them a potent medicinal resource [6,7]. Hence, cassava leaf extraction becomes expedient for efficient cassava utilization.

Plant-derived medicines have gained worldwide popularity in recent years. This is largely due to their safety and efficacy [8]. Atopic dermatitis (AD) is a common chronic recurrent inflammatory skin disease with an increasing incidence in recent decades, especially in developed countries [9]. The current treatment of dermatitis is associated with several limitations, such as the side effects and high recurrence rates of topical glucocorticoids; the potential carcinogenic risks and high costs of topical calcineurin inhibitors; the severe side effects and increased risk of infections associated with systemic immunosuppressants; the high costs and variable efficacy of biologics; and the equipment dependency and long-term risks of phototherapy [10]. Therefore, the search for anti-inflammatory agents from natural plants has become the focus of current research, and the usage of cassava as an anti-inflammatory has been documented in numerous historical prescriptions [11,12]. There has been some prior research on cassava, but it primarily focused on its roots [13]. The cassava root contains 60% to 80% starch, making it a valuable raw material for gluten-free food products, particularly suitable for individuals with gluten intolerance, and it holds potential for development into functional foods [14]. Furthermore, cassava starch is widely used in the production of biodegradable packaging materials owing to its superior film-forming properties and environmentally friendly characteristics. Moreover, the resistant starch in cassava roots functions as a prebiotic, modulating gut microbiota and enhancing intestinal health [15]. However, cassava leaves are abundant in bioactive compounds [16]. Phenolic compounds in cassava leaves exhibit significant antioxidant activity, making them suitable as natural preservatives to extend the shelf life of food products. Flavonoids, such as kaempferol, have been shown to significantly reduce inflammation through the inhibition of the NF-κB signaling pathway [17]. Even though cassava leaves are widely used in many cultures, systematic studies on their chemical composition and biological activity are still limited. Therefore, isolation, identification, and biological activity of cassava secondary metabolites have become crucial for cassava utilization.

In this study, the chemical compositions of cassava leaves were analyzed using liquid chromatography–mass spectrometry. Following this, animal tests were performed to assess the biological activity of green and purple leaves of cassava. This study further investigated the chemical composition of cassava leaves, with their potential uses in food and medicine. Understanding these could form the basis for crop improvement studies, broaden knowledge of tropical plant resources, and advance the health sector.

## 2. Results

### 2.1. Bioactive Metabolites of G.L. and P.L.

To investigate the chemical composition of cassava leaves, two cassava varieties with different colors in leaves were compared in this study. Extractions from G.L. and P.L. were performed using different methods (ET-EA, ET-BU, WE-EA, and WE-BU; see Table 1), and then the products were analyzed via LC–MS. The chemical constituents and proportions of G.L. and P.L. are indicated in Appendix A. In total, 115 and 88 compounds were obtained from the G.L. and P.L., respectively. These bioactive compounds included D-(+)-mannose, trigonelline, rutin, and kaempferol-3-*O*-rutinoside, which possess anti-inflammatory properties [18,19,20].

A series of organic acids (gallic acid, L-pyroglutamic acid, and glutaric acid) and two alkaloid components (betaine and trigonelline) were found. Additionally, terpenoids including ursolic acid, cafestol, lupenone, atractylenolide II, and oleanolic acid were identified. Five flavonoids (rutin, kaempferol-3-*O*-rutinoside, monoglyceride of lauric acid, lupeol, and isoquercitrin) were also derived. Furthermore, astragalin and 12-oxo-phytodienoic acid were unique to G.L., while nicotinamide, apigenin-7-*O*-glucoside (vicenin II), kaempferol, salicylic acid, and the antioxidant ellagic acid were exclusively found in P.L.

### 2.2. Anti-Inflammatory Ability of Bioactive Metabolites in Cassava Leaf Extracts

As indicated in Figure 1, epidermal layer thickness of the mice in the control was modulated, and the tissue structure of their skin layer was normal. Nevertheless, the stained section of the model group (NA) exhibited disorganized dermal tissue, as the structure of dermo–epidermal region was destroyed after histamine application. The epidermal layer was thinner, and the dermal fibers were more clearly aligned in the medication administration group and the positive group (DPH), as compared to the model group. Among these, there was no infiltration of inflammatory cells and a notable improvement in the pathological skin features of the mice in the E group treatment. The results demonstrated the successful establishment of the mouse histamine model. Simultaneously, these results demonstrated that G.L. had a greater effect than P.L. in reducing the inflammatory response brought on by histamine.

Mast cells are immune cells in tissues, responsible for regulating inflammatory and immune responses. To investigate their anti-inflammatory capabilities, we performed toluidine blue staining on mast cells. The blue-purple spots in the tissue represent mast cells (Figure 2). Compared with the model group (NA), the number of mast cells in the treatment groups (A–H) and the positive control group (DPH) decreased. Additionally, the number of mast cells in G.L. was significantly lower than that in P.L. Furthermore, the total number of mast cells was the lowest in the P.L. treated with ET-BU (Group B).

To further confirm the anti-inflammatory effects, the expressions of proinflammatory cytokines were analyzed. As shown in Figure 3, the model group (NA) had higher expression levels of IL-17 and TNF-α than the Control group. However, the expression levels of these cytokines were dramatically reduced following the administration of antihistamines (DPH) and the treatment of cassava leaf extracts. These results indicated that cassava leaf extracts could achieve an anti-inflammatory effect by inhibiting the expression of proinflammatory cytokines. In general, the anti-inflammatory effect of G.L. extract was better than that of P.L., and the effect of water extract of G.L. was the best.

### 2.3. Transcriptomics Analysis

To further characterize the molecular mechanism of the anti-inflammatory effects, transcriptomics analysis of the mice treated with cassava leaf extracts was performed. The degree of difference between the individual samples was analyzed using Principal Component Analysis (PCA), in which the x-axis contributed 26.37% and the y-axis contributed 21.71%, and the degree of aggregation of the samples was compared mainly through the x-axis (Figure 4a). It can be seen that the samples of each experimental group are more aggregated, while the samples of the control group are far away from the experimental group. The results indicate that the experimental group and the control group were effectively separated.

Correlation coefficients (|R| > 0.6) of all samples are presented in Figure 4b,c. The samples of each group had good correlation. The similarity and repeatability between the experimental group and the control group were quite better.

Venn analysis of inter-samples in each group was also performed. Among them, there were 14,693 common genes in groups A, B, C, and D, and 14,724 common genes in groups E, F, G, and H (Figure 5a,c). Subsequently, comparison of the differential gene set was also analyzed between the model group and the experimental group (Figure 5b,d). The gene numbers of B vs. MOD and D vs. MOD were 1684 and 1777, respectively, accounting for 17.28% of the total genes, indicating that the B and D treatments may have similar mechanisms in the treatment of atopic dermatitis. The most differential genes were found in group G vs. MOD, which may be due to more gene regulation during treatment. The common differential genes of G vs. MOD and E vs. MOD accounted for 19.34%, which suggested that they had similar gene regulation of atopic dermatitis.

To identify differentially expressed genes (DEGs), a pairwise comparison of gene expression levels in the volcano map was performed with adjusted *p* value < 0.05 and |logFC| ≥ 1.2 serving as the criteria (Appendix A). Compared to the model group, group A revealed 3077 DEGs (1976 up-regulated, 1101 down-regulated), group B showed 4197 DEGs (2724 up-regulated, 1473 down-regulated), group C had 4275 DEGs (2600 up-regulated, 1675 down-regulated), and group D exhibited 4388 DEGs (2824 up-regulated, 1564 down-regulated). Notably, group D displayed the highest number of differential genes, suggesting a complex genetic profile for this sample compared to the model group. The first three groups share notable up-regulation of fibronectin (*Fn1*) and carboxypeptidase (*Cpz*), while all four groups exhibit up-regulation of mast cell carboxypeptidase (*Cpa3*).

The top ten genes associated with allergic dermatitis and significantly expressed on the volcano map are listed in Appendix A (refer to dermatitis diseases, Allergic Contact gene database, at https://ctdbase.org/, accessed on 18 April 2023). Their regulatory differences are also listed (*p* < 0.05). Among these, *C1qtnf3* belongs to the C1q/TNF-related protein family and is involved in regulating inflammatory responses and metabolic processes; Cma1 is a serine protease that participates in the generation of inflammatory mediators, and inhibiting Cma1 activity can reduce inflammatory responses; *Cpz* may exert anti-inflammatory effects by modulating inflammatory signaling pathways; Fbn1 is an extracellular matrix protein involved in tissue repair and inflammation regulation; *Fn1* exerts anti-inflammatory effects by regulating inflammatory signaling pathways; Mcpt4 is a mast cell-specific protease involved in the release of inflammatory mediators; and *Serpina3c* and *Serpina3n* are involved in regulating the activity of inflammatory mediators, exerting anti-inflammatory effects by inhibiting the generation and release of inflammatory mediators. Lep is an adipokine involved in the regulation of metabolism and inflammatory responses [18,19,20,21,22,23,24].

In comparison to the model group, the numbers of differentially expressed genes in the experimental group (E, F, G, and H) are shown in Appendix A, with group E showing more complex genetic information. Group E detected 4450 DEGs (2909 up-regulated and 1541 down-regulated), Group F had 3217 DEGs (2100 up-regulated and 1117 down-regulated), Group G identified 5489 DEGs (3411 up-regulated and 2078 down-regulated), and Group H revealed 3114 DEGs (2387 up-regulated and 727 down-regulated). Among these differential genes regulation, mast cell chymotrypsin 1 (*Cma1*), polymerin 1(*Mmrn1*), leptin (*Lep*), and mast cell carboxypeptidase (*Cpa3*) had significant differences and appeared more frequently. It is worth mentioning that *Cma1* is stored in the granules of mast cells, and atopic dermatitis is closely related to mast cell activation [25]. Previous studies have demonstrated that inflammation can result in an enhanced expression of *Mmrn1* [26] and *leptin* [27]. Significant regulation of *Cpa3* was similar to that seen in groups A, B, C, and D. However, the up-regulation of *Cma1*, *Mmrn1*, *Lep*, and *Cpa3* appears inconsistent with our observed biological effects. This discrepancy may be attributed to context-dependent mechanisms, such as interactions with anti-inflammatory pathways (e.g., IL-10 or TGF-β), time-dependent roles in inflammation resolution, or post-transcriptional regulation not captured by gene expression data. Further mechanistic studies are required to fully elucidate the reasons behind the up-regulation of these genes.

The top 10 genes associated with allergic dermatitis and significantly expressed on the volcano map are listed in Appendix A (refer to dermatitis diseases, Allergic Contact gene database, at https://ctdbase.org/). Their regulatory differences are also listed (*p* < 0.05). Among these, *Lep* exerts its anti-inflammatory effects by modulating inflammatory signaling pathways such as JAK/STAT; *Adipoq* exerts its anti-inflammatory effects by activating AMPK and inhibiting the NF-κB signaling pathway; *Cfd* is an essential component of the complement system and is involved in inflammatory responses; *Cfh* mitigates inflammatory responses by regulating the complement system; *Ccl21a* exerts its anti-inflammatory effects by modulating immune cell recruitment; *Hp* reduces inflammatory responses by scavenging free hemoglobin; and *Lbp* exerts its anti-inflammatory effects by regulating the TLR4 signaling pathway [28,29,30,31,32,33,34].

In the RNA-seq cluster heat map, expression pattern cluster analysis was performed for the genes in the selected set. Red indicates that the genes had a high expression level in the sample, while blue indicates a low expression level. The left side is the gene clustering tree, and the right side is the gene name. Proximity of gene branches reflects similar expression levels. As shown in Figure 6, there were effective differences in most genes between the experimental group and the control group, and experimental group genes exhibited higher expression than the controls. Nevertheless, expression patterns were consistent within experimental groups, indicating functional correlation among similarly expressed genes.

### 2.4. GO and KEGG Analysis

To investigate the genotype alterations post-pretreatment in each group, GO (Gene Ontology) analysis was performed on screened DEGs, categorizing them into three levels: biological process (BP), molecular function (MF), and cell component (CC). Additionally, KEGG (Kyoto Encyclopedia of Genes and Genomes), an exhaustive database integrating genomic, chemical, and biochemical system data, facilitated a comprehensive understanding of gene functions, interactions, regulatory mechanisms, and metabolic pathways.

In this study, the first five GO terms with *p* < 0.05 significant levels were enriched (Figure 7a,d,h,k and Figure 8a,c,e,g). The KEGG pathway with greater enrichment (*p* < 0.05) and more genes (*n* ≥ 6) was enriched in the experimental group compared to the control group (Figure 7b,e,i,l and Figure 8b,d,f,h).

The significant pathways in group A are ECM–receptor interaction, complement and coagulation cascade, PI3K-Akt signaling pathway, and adhesion plaque. The significant pathways in group B are complement and coagulation cascade, ECM–receptor interaction, hematopoietic cell lineage, and PI3K-Akt signaling pathway. The significant pathways in group C are ECM–receptor interaction, complement and coagulation cascade, PI3K-Akt signaling pathway, and malaria. The significant pathways in group D are complement and coagulation cascade, cytokine–cytokine receptor interaction, and PI3K-Akt signaling pathway. Previous studies have confirmed that inhibiting the proteins on the PI3K-Akt signaling pathway can inhibit mast cell-mediated hypersensitivity [35], and four treatments may have the potential for anti-atopic dermatitis through this coupling pathway. The significant pathways in group E are complement and coagulation cascade, PI3K-Akt signaling pathway, malaria, and cell adhesion molecules. The significant pathways in group F are complement and coagulation cascade, protein digestion and absorption, ECM–receptor interaction, and PI3K-Akt signaling pathway. The significant pathways in group G are PI3K-Akt signaling pathway, complement and coagulation cascade, and ECM–receptor interaction. The significant pathways in group H are complement and coagulation cascade, PI3K-Akt signaling pathway, ECM–receptor interaction, and protein digestion and absorption. It can be seen that G.L., like P.L., may have anti-atopic dermatitis potential through the PI3K-Akt signaling pathway. The common enriched genes of all samples from GO term and the KEGG path analyses are listed in Appendix A.

The numbers of up-regulated and down-regulated genes annotated to certain pathways and their classifications are shown in Figure 7c,f,j. In each classification, human diseases contain the most genes, followed by biological systems. The signaling pathways with the largest up/down-regulation differences in these groups were signal transduction and the immune system.

### 2.5. Network Pharmacology Analysis

The main components of cassava leaf extract, including D-(+)-mannose, rutin, kaempferol-3-*O*-rutinoside, oleamide, and 1-stearoylglycerol, were input into the PubChem website (https://pubchem.ncbi.nlm.nih.gov/, accessed on 18 April 2023) to obtain their isomeric SMILES structures. These structures were then uploaded to the Swiss Target Prediction database (http://www.swisstargetprediction.ch/, accessed on 18 April 2023), with the species selected as “Homo sapiens”. Targets were filtered using a probability threshold of >0.1, resulting in the prediction of 100 potential targets, which were exported in CSV format. Next, using the keywords “atopic dermatitis” and “skin repair”, the GeneCards database (GeneCards—Human Genes | Gene Database | Gene Search) was searched to identify targets related to skin inflammation. A total of 12,480 targets were enriched and exported in CSV format. The targets predicted from cassava leaf extract and the skin repair-related targets were then imported into the Venny 2.1.0 tool (Venny 2.1.0 (csic.es)) to identify overlapping genes. This analysis yielded 92 core gene targets, as illustrated in Figure 9a. The shared protein genes were uploaded to the STRING website, generating a preliminary protein–protein interaction (PPI) network diagram (Figure 9b). Subsequently, using Cytoscape 3.7.2 software, a compound–target interaction network was constructed. Core nodes were screened based on network topological features, such as node degree, and the resulting network diagrams are shown in Figure 9c,d. For each gene, its fundamental functions are determined by its protein domains and the research literature. Gene Ontology (GO) and Kyoto Encyclopedia of Genes and Genomes (KEGG) are databases that store gene-related functions based on different classification principles. Using the DAVID website (https://david.ncifcrf.gov/, accessed on 18 April 2023), GO enrichment analysis was performed, and data with *p* < 0.01 were screened, resulting in a total of 59 GO terms. These included 20 BP, 18 CC, and 21 MF. To better visualize the gene functions, the terms were sorted based on *p*-value and count, and a bar chart was generated using the bioinformatics website (https://www.bioinformatics.com.cn/, accessed on 18 April 2023) (Figure 9e). Analysis of biological processes revealed that the key target genes were primarily involved in the negative regulation of apoptotic processes, negative regulation of lipid catabolic processes, negative regulation of inflammatory responses, positive regulation of nitric oxide biosynthetic processes, and positive regulation of MAPK cascades. Analysis of CC indicated associations with spindle microtubules, brush border membranes, GABA-ergic synapses, postsynapses, postsynaptic membranes, basolateral plasma membranes, neuronal cell bodies, and glutamatergic synapses. MF terms included carbonate dehydratase activity, 3′,5′-cyclic-nucleotide phosphodiesterase activity, long-chain fatty acid transmembrane transporter activity, histone demethylase activity, 3′,5′-cyclic-AMP phosphodiesterase activity, 3′,5′-cyclic-GMP phosphodiesterase activity, protein serine/threonine/tyrosine kinase activity, dioxygenase activity, steroid binding, and fatty acid binding. KEGG pathway analysis based on the DAVID database identified a total of 19 signaling pathways (Figure 9f). By ranking the pathways according to -Log *p* values and the number of enriched genes, it was found that cassava leaf extract primarily participates in potential pathways related to alleviating skin inflammation, including the PPAR signaling pathway, PI3K-Akt signaling pathway, cGMP-PKG signaling pathway, insulin signaling pathway, Fc epsilon RI signaling pathway, and steroid hormone biosynthesis [36,37,38,39,40,41]. The key targets involved in these pathways include receptor families (nuclear receptors, receptor tyrosine kinases, G protein-coupled receptors, etc.), kinase families (PI3K, Akt, mTOR, Syk, PLCγ, etc.), transcription factor families (NF-κB, FOXO, CREB, etc.), metabolic enzyme families (GLUT4, AMPK, CYP enzymes, etc.), and cytokine and ion channel families (IL-4, IL-10, K^+^ channels, etc.).

### 2.6. Molecular Docking Analysis

CDOCKER ENERGY reflected the overall docking energy based on the three-dimensional structure and physicochemical properties of both the ligand and the protein [42]. It quantified the extent to which intermolecular forces—such as van der Waals forces, electrostatic interactions, and hydrogen bonds—contributed to the total binding affinity. It was hypothesized that ligand–receptor complexes exhibiting lower CDOCKER ENERGY values demonstrated robust interactions with the receptor. At the time, there was no universal standard for the screening of active molecular targets. Figure 10 illustrates six docking results involving three target proteins and two primary compounds. Table 1 depicts the interaction energies of these three target proteins with the two main compounds. The molecular docking results revealed that the affinities of the identified three target proteins and two primary compounds were all below −15.0 kcal/mol, suggesting that the ligands and receptors exhibited strong binding activity. This indicated that the two compounds might have exerted anti-inflammatory effects by acting on the three inflammation-associated proteins.

## 3. Discussion

Cassava thrives in a wide range of climatic conditions and has high adaptability to high temperatures, drought, salinity, and low fertility soils [7,43,44]. Cassava leaves are abundant in various nutrients, including protein, vitamin A, β-carotene, as well as minerals. Liquid chromatography–mass spectrometry was performed to analyze the chemical components of cassava leaves, and animal experiments were conducted to evaluate the bioactivities of green (G.L.) and purple (P.L.) leaves of cassava. In this study, cassava leaf extracts were found to be rich in bioactive metabolites such as D-(+)-mannose, trigonelline, rutin, kaempferol-3-*O*-rutinoside, and oleamide. D-(+)-mannose [18,45], trigonelline [19,21], rutin [28], kaempferol-3-*O*-rutinoside [20], and oleamide [23,24] were previously reported with the anti-inflammatory activity. The primary component of cassava roots is starch, whereas cassava leaves are predominantly composed of protein and have more active substances [29]. However, leaves typically contain higher levels of cyanogenic compounds compared to roots. In our chemical analysis, cyanogenic compounds were not detected, which suggests that the extraction method employed may have selectively favored the extraction of bioactive compounds while effectively excluding toxic substances.

To explore and compare the anti-inflammatory and immunomodulatory potential of active components extracted from leaves of different cassava varieties using different extraction methods, an inflammatory response was induced in mice by stimulating them with histamine. Histamine strongly stimulates vascular endothelial cells, leading to vasodilation and increased permeability. Epidermal hyperplasia is a characteristic of skin damage and is commonly used to evaluate the inhibitory effect of drugs on epidermal hyperplasia caused by allergic reactions [30]. As shown by histological staining and immunohistochemical results, under LPS stimulation, the epidermal layer of mice significantly thickened, and the number of mast cells in the deeper layers of the skin markedly increased. Simultaneously, immune cells in the skin were activated, such as macrophages and dendritic cells, inducing these cells to produce large amounts of TNF-α and IL-17. TNF-α is a molecular form of the tumor necrosis factor family, which significantly increases under pathological conditions such as infection, inflammation, and tumors. TNF-α exhibits various biological activities, including immunomodulation [46], inflammation mediation [47], and antitumor effects, and is also involved in tissue repair, metabolic regulation, and physiological sleep regulation. IL-17 is mainly secreted by immune cells in the skin, such as γδT cells, innate lymphoid cells, and CD4^+^ T cells. During skin inflammation, the number of these immune cells significantly increases, and the levels of IL-17 were highly expressed. High levels of IL-17 can induce the production of other inflammatory cytokines, such as TNF-α and IL-1, thereby exacerbating the skin’s inflammatory response and deepening skin inflammation [48]. Additionally, IL-17 is involved in regulating the skin’s immune response and barrier function. During inflammation, IL-17 may affect the proliferation, differentiation, and apoptosis of skin cells, leading to impaired skin barrier function and exacerbating skin dryness and aging.

Although different cassava leaf extracts exhibited therapeutic effects on skin inflammation, there were differences in the repair effects of the two types of cassava leaf extracts and different extraction methods from the same cassava leaves on inflammation. The ET-EA group showed better inhibition of abnormal proliferation and differentiation of local skin cells. This may be due to the higher content of diethanolamine, 12-oxo-phytodienoic acid (OPDA), monolaurin, corsolic acid, ethyl linolenate, and 1-linoleoyl-glycerol after ethanol and *n*-butanol treatment. These active ingredients have been studied in the cosmetics and skin care industry for their unique physiological effects. OPDA, an anti-inflammatory compound derived from plant lipids, inhibits the expression of typical inflammatory cytokines interleukin-6 and TNF-α induced by LPS by inhibiting inflammatory signaling [48]. Monolaurin regulates avian cell apoptosis and inflammation by inhibiting LPS-induced ROS production and NF-κB activation [49]. Corsolic acid, a pentacyclic triterpenoid compound, not only has significant anti-inflammatory effects and inhibits inflammatory responses but also exhibits antioxidant and other various biological activities [50]. Ethyl linolenate, as an ethyl ester of ω-3 unsaturated fatty acids, not only has anti-inflammatory effects [51] but also moisturizes and repairs the skin barrier, improves skin texture, and exhibits certain antioxidant activity [32]. 1-Linoleoyl-glycerol, as the glyceride form of linoleic acid, also has anti-inflammatory, moisturizing, and skin barrier-repairing effects, benefiting skin health [52]. The application of these active ingredients in cosmetics provides strong support for improving skin inflammation and promoting skin health. Furthermore, the extracts of P.L. obtained through water extraction followed by ethyl acetate extraction exhibited better inhibitory effects on IL-17 and TNF-α. We speculate that this processing method can retain more triethanolamine, DL-lactic acid [53], L-pyroglutamic acid [54,55,56], and bis (4-ethylbenzylidene) sorbitol [57].

KEGG results indicate that two varieties of cassava leaf extracts participate in the treatment of inflammatory responses through the ECM–receptor interaction, PI3K-Akt signaling pathway, and complement and coagulation cascade pathways. The complement and coagulation cascade is an important defense mechanism for the body to respond to foreign pathogen invasion and maintain tissue homeostasis. In this study, extracts of G.L. and P.L may inhibit the structural and functional disruption of the ECM by inflammatory mediators, affecting the interaction between cells and the ECM. Furthermore, ECM–receptor interactions regulate cell proliferation, migration, and differentiation processes, impacting tissue repair and regeneration.

Network pharmacology analysis identified 92 core gene targets shared between cassava extract and skin repair processes, which were enriched in pathways such as PPAR, PI3K-Akt, and cGMP-PKG signaling. GO and KEGG analyses highlighted their roles in regulating apoptosis, lipid metabolism, inflammation, and nitric oxide biosynthesis, with key targets including receptor families, kinases, transcription factors, and metabolic enzymes. The molecular docking analysis revealed that cassava leaf extract components, particularly 1-stearoylglycerol and oleamide, exhibited strong binding affinities with inflammation-associated proteins (AKT1, TNF, BRAF), suggesting potent anti-inflammatory effects. These findings indicated that cassava leaf extract modulated multiple pathways to alleviate skin inflammation and promote repair, supported by robust computational evidence.

While this study has focused on the chemical composition and bioactive effects of cassava leaves, revealing their potential anti-inflammatory properties, several limitations should be acknowledged. First, the study only examined two specific cassava leaf varieties, which may not fully represent the chemical profiles and bioactive potentials of other cultivars. Second, although transcriptomic analyses suggested that the anti-inflammatory effects of cassava leaf extracts may be related to the modulation of mast cell activation-related genes (e.g., *Cma1*, *Cpa3*, and *Fn1*), the exact molecular mechanisms and pathways involved remain to be fully elucidated. Furthermore, since the modeling procedure involves abdominal injection, whereas drug administration is performed via topical application on the back, the distinct drug delivery routes may result in variations in drug absorption and distribution, which could potentially influence the experimental outcomes [33]. Additionally, the study did not address the safety and toxicity profiles of cassava leaf extracts, which are critical for their practical application as natural anti-inflammatory agents. While our results demonstrated significant anti-inflammatory effects, we acknowledge that antioxidant activity testing could provide further insights into the mechanisms underlying these effects. Antioxidant activity is often closely associated with anti-inflammatory properties, as reactive oxygen species (ROS) play a critical role in the inflammatory response. By neutralizing ROS, antioxidants can mitigate oxidative stress and, consequently, inflammation [34]. Moreover, we observed that the therapeutic efficacy of the positive control drug in pathological studies appeared to be less than optimal. We postulate that this may be attributable to a combination of factors, including individual variability among mice and an insufficient sample size. Despite these challenges, we endeavored to ensure the reliability and validity of our results through meticulous experimental design, randomization, rigorous statistical analysis, and the inclusion of control groups. Although the cassava leaf extract demonstrated significant biological activity, our objective was not to surpass the positive control drug but rather to evaluate its potential as an alternative therapeutic agent. Future research should aim to further explore the underlying mechanisms of action to fully realize the therapeutic potential of cassava leaf extracts.

## 4. Materials and Methods

### 4.1. Plant and Animal Materials

Plants: Two cassava varieties, South China No. 9 and South China No. 20, characterized by their green (G.L.) and purple leaves (P.L.), respectively, were cultivated at the National Cassava Germplasm Repository (NCGR) at Danzhou, Hainan, Province, China (19°30′ N, 109°30′ E). Animals: Male Kunming mice (5 weeks of age, 29–33 g) used in this study were obtained from the Guangdong Laboratory Animal 95 Center (SCXK/20130002). Table 2 lists information about plant collection and extraction procedures.

### 4.2. The Extraction of Cassava Leaves

The leaves of both varieties SC9 (G.L.) and SC20 (P.L.) were harvested. Next, the leaves were cleaned, air-dried, and crushed into fine powders. The dried powder was discretely sealed in bags and stored at room temperature until use.

Ethanol Extraction (ET): A mass of 250 g of cassava leaf powder was mixed with 95% ethanol at a ratio of 1:10 (*w*/*v*). The mixture was heated at 50 °C for 3 h. After heating, the mixture was filtered to remove the powder, and vacuum rotary evaporation was performed at 50 °C to remove the ethanol. Once the ethanol was removed, 100 mL of water was added to dissolve the residue, and the solution was divided into two parts. To one part, an equal volume of ethyl acetate (EA) was added, and to the other part, an equal volume of *n*-butanol (BU) was added. The mixtures were allowed to stand for phase separation, and the ethyl acetate and *n*-butanol extracts were collected separately. Vacuum rotary evaporation was used again to remove the ethyl acetate and *n*-butanol, yielding the extracts. The extracts were dissolved in methanol at a concentration of 1 mg/mL. Finally, the solution was filtered through a 0.22 µm membrane.

Water Extraction (WE): Cassava leaf powder (250 g) was suspended in water (1:15 *w*/*v*), heated at 50 °C for 3 h, and then processed according to the ethanol extraction method. The extract was sectioned into two, and equal proportions of ethyl acetate and *n*-butanol were discretely added. The mixture was allowed to stand for 24 h and concentrated using a rotary evaporator. Next, the mixture was dissolved in 1 mg/mL methanol and filtered through a 0.22 μm filter membrane. A total of eight extracts obtained were stored at 4 °C (Table 3).

### 4.3. UPLC-Q-TOF/MS

The LC–MS/MS system utilized was a Thermo Scientific Ultimate 3000 liquid chromatography system (Thermo Fisher Scientific, Waltham, MA, USA), integrated with a Q Exactive Orbitrap mass spectrometer (Thermo Fisher Scientific, Waltham, MA, USA) and equipped with an electrospray ionization (ESI) source. A 2 μL sample volume was injected onto a Hypersil Gold C18 column (100 × 2.1 mm, 1.9 μm, Thermo Scientific, Waltham, MA, USA) maintained at a temperature of 40 °C. The liquid chromatography (LC) flow rate was set to 250 μL/min, employing a mobile phase consisting of water (containing 0.1% formic acid) (A) and methanol (B). The gradient elution program was executed as follows: from 0 to 5 min, 95% solvent A; from 5 to 20 min, a linear decrease from 95% to 60% A; from 20 to 30 min, a further decrease to 10% A; from 30 to 35 min, maintained at 10% A; from 35 to 37 min, a ramp back to 95% A; and from 37 to 45 min, held at 95% A.

Both positive and negative ESI modes were employed to detect the MS signals of the analytes, with spray voltages set at +3.5 kV and −2.5 kV, respectively. The sheath gas flow rate, auxiliary gas flow rate, and sweep gas flow rate were adjusted to 40, 10, and 0 arbitrary units, respectively. The capillary temperature was maintained at 320 °C, while the auxiliary gas heater temperature was set to 350 °C. The instrument was programmed to alternate between positive and negative ion scanning modes. The scan mode was configured for full MS scan-dd MS2, acquiring initial MS signals at a resolution of 70,000 fwhm, and targeted MS/MS scans were conducted at a resolution of 17,500 fwhm with an isolation width of 0.4 m/z. The m/z scan range was set from 50 to 750. Instrument control and data acquisition were facilitated through an Xcalibur workstation (Thermo Fisher Scientific, Waltham, MA, USA). Data analysis was performed using Compound Discoverer 3.2 (Thermo Scientific, Waltham, MA, USA) and the mzCloud database (Thermo Scientific, Waltham, MA, USA, http://www.mzcloud.org, accessed on 18 April 2023), which may be accessed for further compound identification and characterization. Please note that access to the mzCloud database may be subject to availability and network conditions.

### 4.4. Modeling and Drug Delivery

An animal modeling and drug delivery system were presented in this study. The research was conducted following the regulations of the People’s Republic of China on the Administration of Experimental Animals. After a week of acclimatization, 44 mice were randomly assigned to eleven groups (*n* = 4 per group), which comprised one blank control group (Control), one histamine model control group (NA), one antihistamine-positive control group (DPH), and eight cassava leaf extract-treated groups. The samples were kept under controlled conditions: temperature (26 °C), relative humidity (40–70%), and a 12-hour light–dark cycle.

The mice had a 2 cm × 3 cm area of fur shaved from their neck and back two days before the experiment. Except for the control group, mice in other groups were intradermally injected with 500 μg (50 μL) of histamine on the 1st, 4th, and 7th days after treatment. The model was successfully constructed when the mice showed skin redness and itching symptoms. After eight days, 100 μL (4%) of the cassava leaf extracts was applied daily to the shaved area of the treated groups. Meanwhile, the positive control group received 100 μL of DPH (0.5 mg/mL) for 14 days, following the previous report [58]. Histamine injections were further administered at 10, 13, 16, and 19 days. One hour after the final treatment, the mice were euthanized by cervical dislocation.

### 4.5. Hematoxylin and Eosin (HE) Staining 

The mouse skin tissue was fixed in 4% paraformaldehyde for more than 24 h, with the dehydration box placed in a gradient alcohol and xylene solution for thorough dehydration. The dehydrated tissue was embedded in a melted wax block, sliced with a paraffin microtome to 4 μm slices, and oven-dried at 60 °C. Next, the paraffin tissue was deparaffinized with xylene solution and gradient alcohol and rinsed with distilled water. After rinsing with tap water, the slices were stained with a hematoxylin dye solution for 3–5 min. This was followed by sequential rinsing with tap water, differentiation, and staining with blue solution. The slices were then rinsed with running water. The sliced sections were further dehydrated with 85% gradient alcohol for 5 min, and then stained with eosin solution for another 5 min. The dehydrated slices sections were then immersed in xylene for a transparent appearance, before being sealed with neutral gum (Sinopharm Group Chemical Reagent Co., Ltd., Shanghai, China).

Histological changes were observed under a light microscope (Olympus Co., Tokyo, Japan), and epidermal thickness was measured using IMAGE Pro PLUS software (version 6.0, DMi8, LEICA, Wetzlar, Germany).

### 4.6. Toluidine Blue (TB) Staining

After the paraffin sections were dried, they were deparaffinized with xylene solution and gradient alcohol, rinsed with distilled water, incubated at 60 °C, and stained with toluidine blue (1%) for 40 min. Following this, the stained section was rinsed with distilled water, dehydrated in gradient alcohol, immersed in xylene for transparent appearance, and finally sealed with natural gum. Subsequently, a fluorescence microscope (DMi8, LEICA, Wetzlar, Germany) were used for microscopy analysis.

### 4.7. Immunohistochemistry (IHC) Staining

After being deparaffinized and hydrated, paraffin sections were repaired in a box filled with citric acid antigen retrieval buffer (pH = 6.0). The repaired section was then incubated with 3% hydrogen peroxide solution to block endogenous peroxidase. Subsequently, the section was rinsed, blocked with serum, and mixed with primary and secondary antibodies before incubation. Next, diaminobenzidine (DAB) was used for color development, nuclei were counterstained with hematoxylin solution, and the slides were dehydrated.

The indexes measured in the experiment were IL-17 and TNF-α. The measured index is based on the blank group’s normal index and the model group’s outlier index. After medication, the change trend of each index was observed at intervals, and the actual influence trend was recorded.

### 4.8. Transcriptomics Analysis

The RNA from skin tissues was extracted using TRIzol reagent. Each RNA sample was sent to Shanghai Meiji Biomedical Technology Co., Ltd. for transcriptomics analysis using the Illumina HiSeq 4000 platform. Quality control analysis was performed to remove low-quality reads and adapter sequences. Further quality control analysis filtered out raw reads above 10% ambiguous bases (N), 50% low-quality bases (Q ≤ 20), and excessive base A. The Bowtie2 tool was used for ribosome alignment, and HISAT2 software was employed for sequence and NCBI reference genome alignment. Quantification of gene expression levels was performed across all samples using StringTie software. Correlation analysis was performed using R (version 4.0.0). Differential gene expression analysis was conducted using DESeq software (version 1.26.0), a component of the DEGseq package in the R language. Genes screened with an adjusted *p*-value (FDR) < 0.05 and |log2(fold change)| > 1 were considered significantly differentially expressed. GO terms and KEGG enrichment analysis were analyzed to explain the molecular function and pathways regulating the differentially expressed genes.

### 4.9. Network Pharmacology Analysis

The relevant targets were identified by searching the GeneCards database (www.GeneCards.org/, accessed on 18 April 2023) using the keywords “atopic dermatitis” and “skin repair”. The potential targets of differential metabolites (DMs) were extracted from the SEA (https://sea.bkslab.org/, accessed on 18 April 2023) and TCMSP databases (http://tcmspw.com/tcmsp.php, accessed on 18 April 2023), and their intersections with atopic dermatitis and skin repair were analyzed using Venny (https://bioinfogp.cnb.csic.es/tools/venny/index.html, accessed on 18 April 2023). The STRING tool (www.string-db.org/, accessed on 18 April 2023) was employed to investigate direct and indirect interactions among these targets. Subsequently, a preliminary PPI network map was constructed to depict the key targets. The DAVID tool (https://david.ncifcrf.gov, accessed on 18 April 2023) was utilized to categorize GO terms and enriched pathways in the KEGG. The visualization of GO terms and KEGG pathways was achieved using the online platform Weishengxin (https://www.bioinformatics.com.cn/, accessed on 18 April 2023).

### 4.10. Molecular Docking Analysis

The three-dimensional structures of the small molecular ligands, 1-stearoylglycerol and oleamide, present in cassava leaf extracts, were obtained from PubChem (https://pubchem.ncbi.nlm.nih.gov/, accessed on 18 April 2023), while protein structures were retrieved from the PDB database (RCSB PDB: Homepage) (AKT1: 5wbl; TNF: 2az5; BRAF: 5ct7). The ligand was optimized using the Prepare Ligand module in Discovery Studio (version 19.1.0) with the default settings. The CDOCKER algorithm was utilized to determine the proteins’ active sites, employing default settings. A total of 10 runs were conducted, and the optimal binding pose for each ligand was selected according to the CDOCKER interaction energy.

### 4.11. Statistical Analysis

The composition was determined by low-resolution mass spectrometry using Xcalibur 4.1 software (Thermo Fisher Inc.). The accurate molecular weight values of the measured compounds were matched with the theoretical values and searched in Compound Discoverer’s mz-Cloud database for analysis and identification. The mass deviation of the predicted compounds was controlled in 5 ppm.

Statistical analysis was conducted using GraphPad Prism software 8.3. One-way analysis of variance was used to determine significant difference at the *p* < 0.05 and *p* < 0.01 levels.

## 5. Conclusions

A comprehensive analysis of chemical components of G.L. and P.L. was conducted under different extraction methods using UPLC-Q-TOF/MS technology in this study. Cassava leaf extracts are rich in various bioactive metabolites, including D-(+)-mannose, trigonelline, rutin, kaempferol-3-*O*-rutinoside, and oleamide, which exhibited notable anti-inflammatory properties. Comparative studies reveal that the anti-inflammatory activity of G.L. was superior to that of P.L. Ethanol extraction followed by ethyl acetate extraction is the best extraction method for G.L. Water extraction followed by *n*-butanol extraction is the best extraction method for P.L. Animal models were established to evaluate the anti-inflammatory efficacy of these bioactive metabolites. Compared to the histamine group (NA), the group treated with cassava leaf extracts exhibited reduced epidermal thickness in HE staining, a significant decrease in the number of mast cells in T.B. staining, and lowered expression levels of inflammatory cytokines (IL-17 and TNF-α) in immunohistochemical experiments. Transcriptomic analysis further suggested that the anti-inflammatory effects of cassava leaves may be related to the modulation of genes involved in mast cell activation, such as *Cma1*, *Cpa3*, and *Fn1*. Network pharmacology analysis revealed that cassava leaf extract targets key biological processes, cellular components, and molecular functions, including the regulation of apoptosis, inflammation, and metabolic pathways. The extract was implicated in skin inflammation alleviation through multiple signaling pathways, such as PPAR, PI3K-Akt, and cGMP-PKG, involving diverse targets like receptors, kinases, transcription factors, and cytokines, highlighting its multifaceted therapeutic potential. The molecular docking results demonstrated strong binding activity of 1-stearoylglycerol and oleamide, derived from cassava extracts, with three inflammation-associated proteins: AKT1, TNF, and BRAF. These findings provide new insights into our understanding of the mechanisms underlying the anti-inflammatory effects of cassava leaves. Therefore, cassava leaf extracts exhibit great potential as natural anti-inflammatory agents. Future research can further explore the possibility of developing anti-inflammatory drugs or health supplements based on cassava leaf extracts.

## Figures and Tables

**Figure 1 ijms-26-04140-f001:**
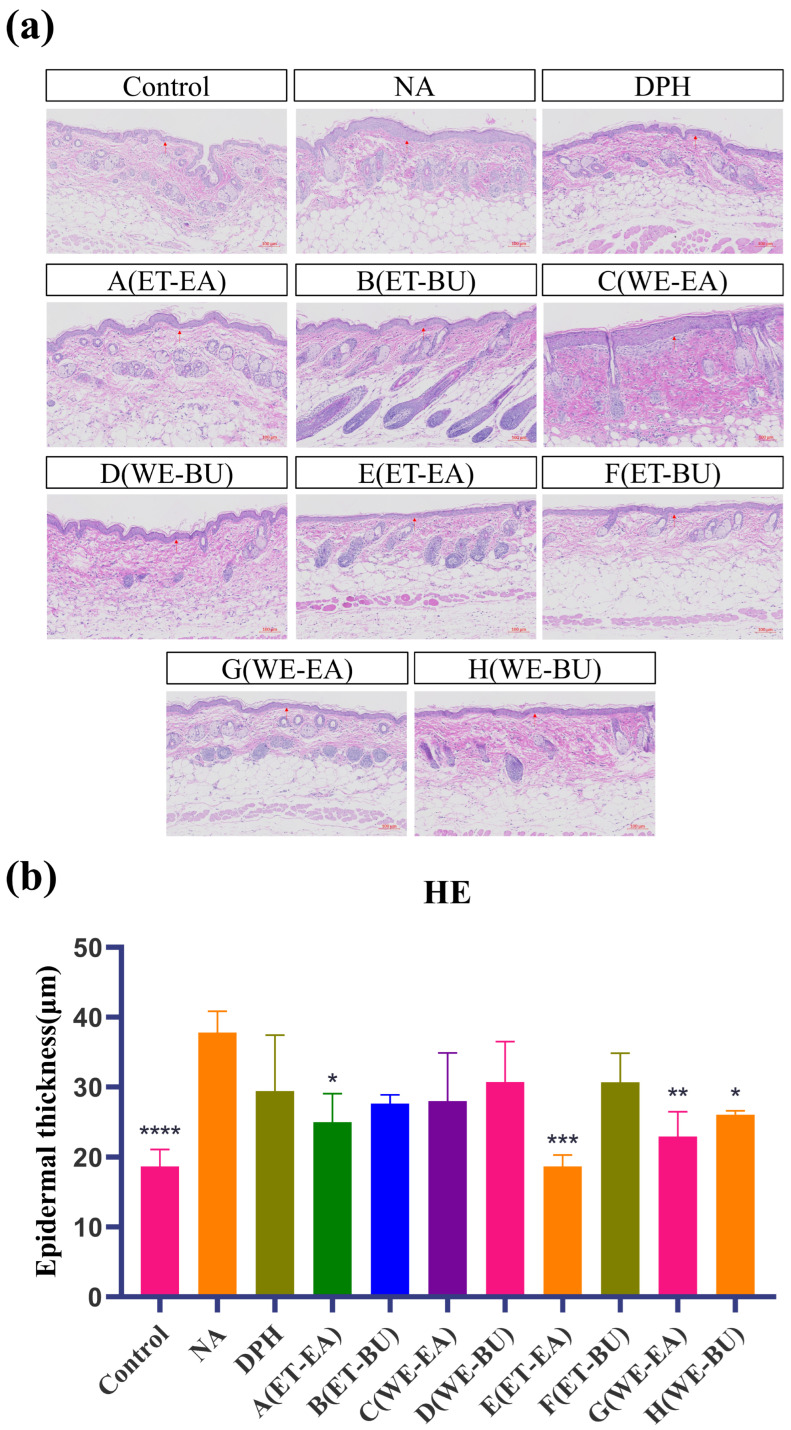
(**a**) Hematoxylin–eosin-stained skin epidermal thickness puzzle (scale bar = 100 μm; *n* = 4). (**b**) Hematoxylin–eosin-stained mice skin epidermal thickness histogram (* *p* < 0.05, ** *p* < 0.01, *** *p* < 0.001, **** *p* < 0.0001). **Note**: Control represents the blank group; NA represents the histamine model group; DPH represents the antihistamine positive group; A (ET-EA) represents the ethanol extraction followed by ethyl acetate extraction group of ziyehuangxin leaves (P.L.); B (ET-BU) represents the ethanol extraction followed by *n*-butanol extraction group of ziyehuangxin leaves; C (WE-EA) represents the water extraction followed by ethyl acetate extraction group of ziyehuangxin leaves; D (WE-BU) stands for water extraction followed by *n*-butanol extraction group of ziyehuangxin leaves; E (ET-EA) stands for ethanol extraction followed by *n*-butanol extraction group of SC9 leaves; F (ET-BU) stands for ethanol extraction followed by *n*-butanol extraction group of SC9 leaves; G (WE-EA) stands for water extraction followed by ethyl acetate extraction group of SC9 leaves; and H (WE-BU) stands for water extraction followed by *n*-butanol extraction group from SC9 leaves.

**Figure 2 ijms-26-04140-f002:**
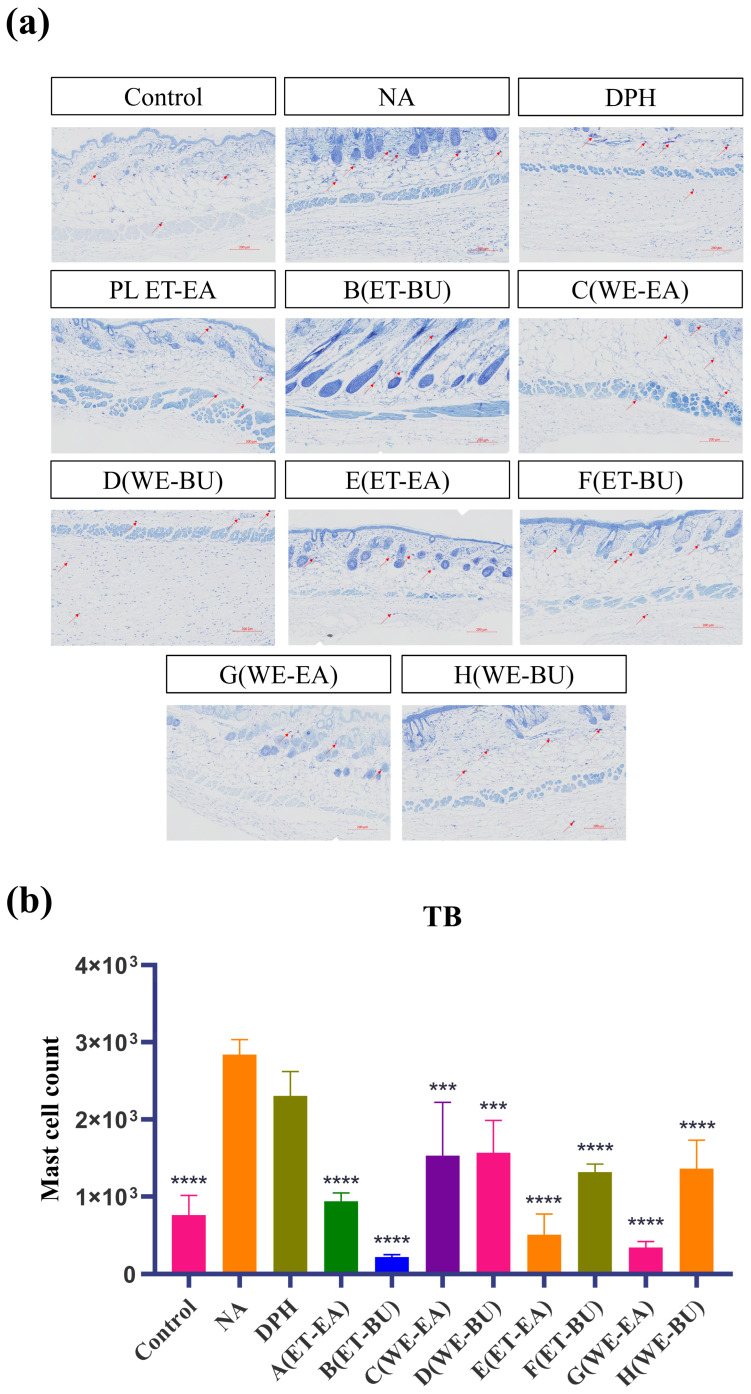
(**a**) Nissl staining showing the total number of mast cells and degranulated cells puzzle (TB, 10×) (scale bar = 200 μm; *n* = 4); (**b**) Nissl staining showing the total number of mast cells and degranulated cells histogram (*** *p* < 0.001, **** *p* < 0.0001). **Note**: Control represents the blank group; NA represents the histamine model group; DPH represents the antihistamine positive group; A (ET-EA) represents the ethanol extraction followed by ethyl acetate extraction group of ziyehuangxin leaves (PL); B (ET-BU) represents the ethanol extraction followed by *n*-butanol extraction group of ziyehuangxin leaves; C (WE-EA) represents the water extraction followed by ethyl acetate extraction group of ziyehuangxin leaves; D (WE-BU) stands for water extraction followed by n-butanol extraction group of ziyehuangxin leaves; E (ET-EA) stands for ethanol extraction followed by *n*-butanol extraction group of SC9 leaves; F (ET-BU) stands for ethanol extraction followed by *n*-butanol extraction group of SC9 leaves; G (WE-EA) stands for water extraction followed by ethyl acetate extraction group of SC9 leaves; and H (WE-BU) stands for water extraction followed by *n*-butanol extraction group from SC9 leaves.

**Figure 3 ijms-26-04140-f003:**
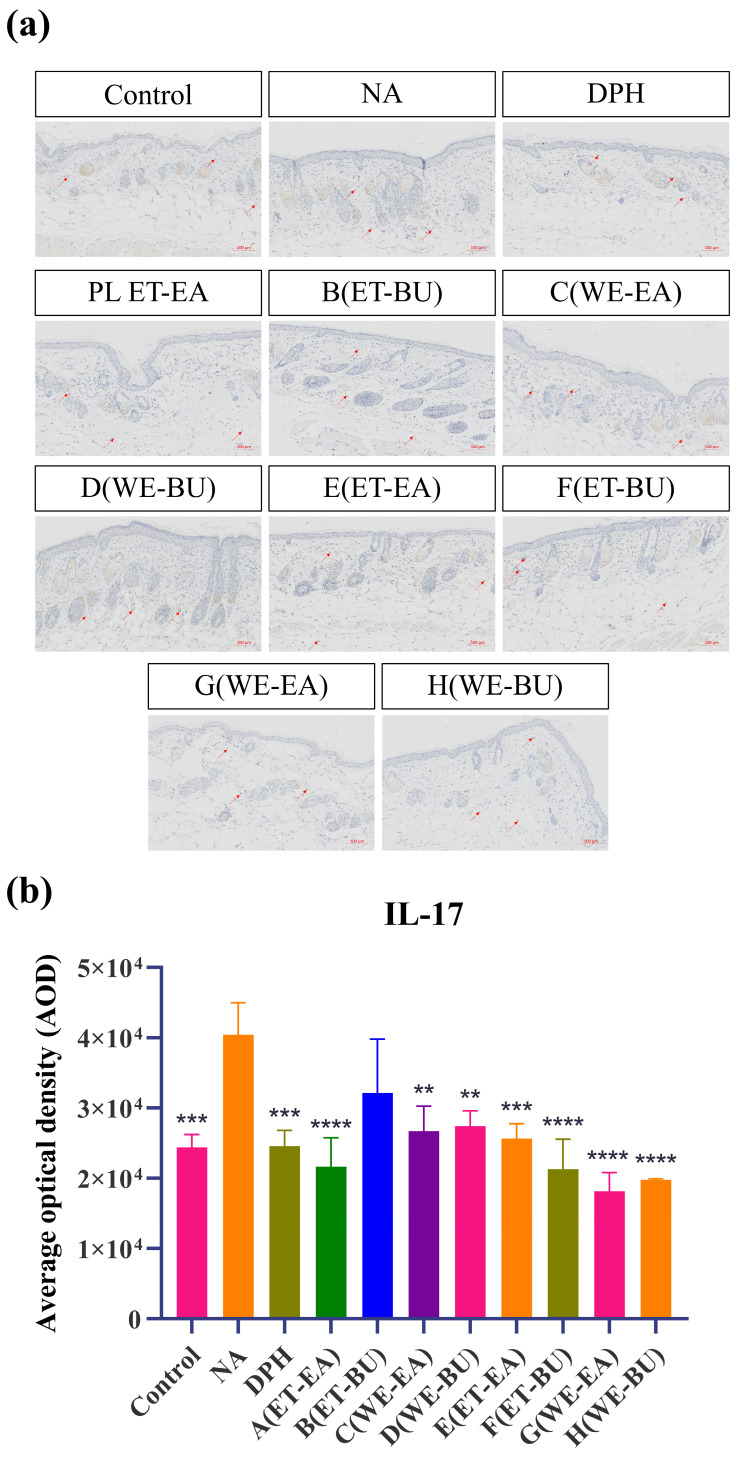
(**a**) IL-17 expression puzzle (scale bar = 100 μm; *n* = 4). (**b**) IL-17 expression histogram (** *p* < 0.01, *** *p* < 0.001, **** *p* < 0.0001). (**c**) TNF-α expression puzzle (scale bar = 100 μm; *n* = 4). (**d**) TNF-α expression histogram (*** *p* < 0.001, **** *p* < 0.0001). Note: Control represents the blank group; NA represents the histamine model group; DPH represents the antihistamine positive group; A (ET-EA) represents the ethanol extraction followed by ethyl acetate extraction group of ziyehuangxin leaves (PL); B (ET-BU) represents the ethanol extraction followed by *n*-butanol extraction group of ziyehuangxin leaves; C (WE-EA) represents the water extraction followed by ethyl acetate extraction group of ziyehuangxin leaves; D (WE-BU) stands for water extraction followed by *n*-butanol extraction group of ziyehuangxin leaves; E (ET-EA) stands for ethanol extraction followed by *n*-butanol extraction group of SC9 leaves; F (ET-BU) stands for ethanol extraction followed by *n*-butanol extraction group of SC9 leaves; G (WE-EA) stands for water extraction followed by ethyl acetate extraction group of SC9 leaves; and H (WE-BU) stands for water extraction followed by *n*-butanol extraction group from SC9 leaves.

**Figure 4 ijms-26-04140-f004:**
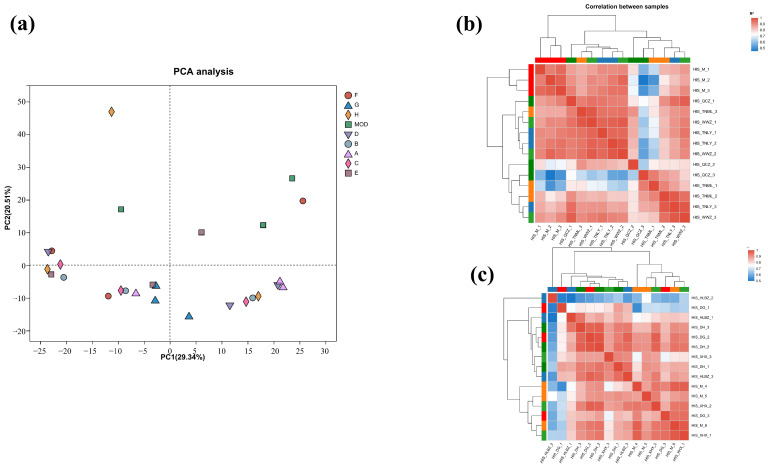
(**a**) PCA diagram; (**b**) sample correlation coefficient diagram for groups A, B, C, and D; (**c**) sample correlation coefficient diagram of groups E, F, G, and H.

**Figure 5 ijms-26-04140-f005:**
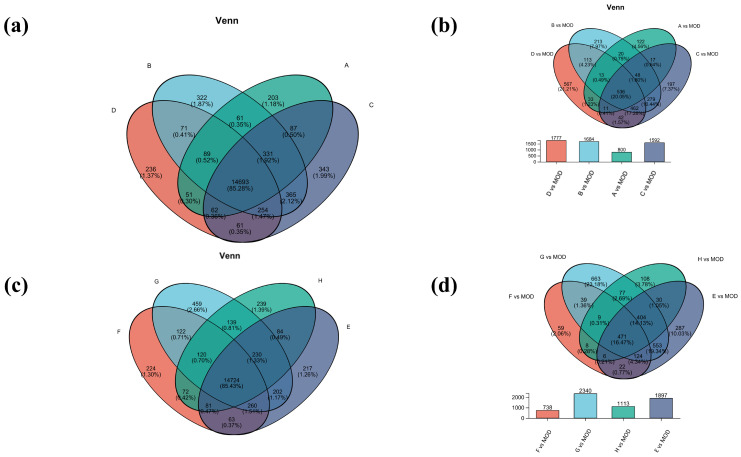
Venn analysis diagram. (**a**) Venn analysis of groups A, B, C, and D; (**b**) differential gene sets for groups A, B, C, and D; (**c**) Venn analysis of groups E, F, G, and H; (**d**) differential gene sets for groups E, F, G, and H.

**Figure 6 ijms-26-04140-f006:**
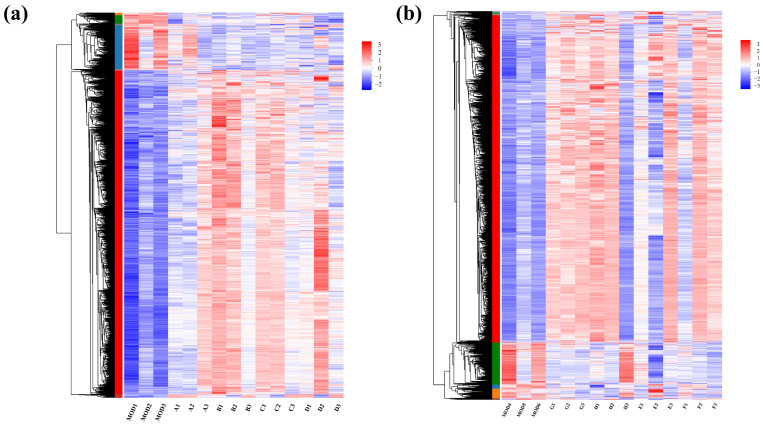
Clustering heat maps for (**a**) groups A, B, C, and D and (**b**) groups E, F, G, and H.

**Figure 7 ijms-26-04140-f007:**
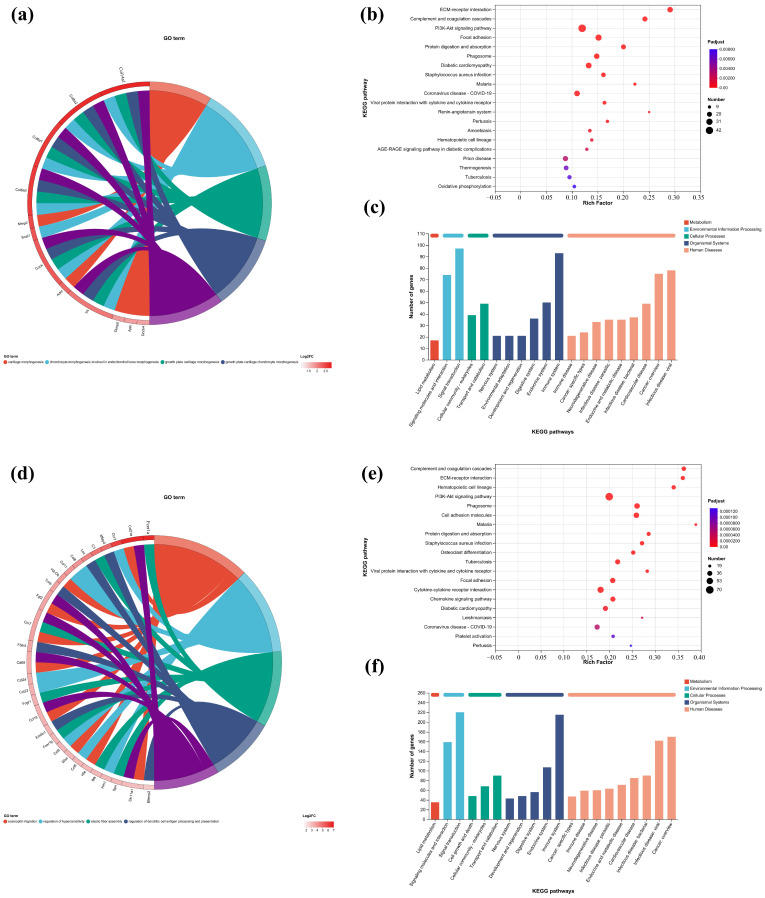
GO enrichment chords of (**a**–**c**) P.L. ET-EA (A group), (**d**–**f**) P.L. ET-BU (B group), (**g**–**i**) P.L. WE-EA (C group), (**j**–**l**) P.L. WE-BU (D group), enrichment of KEGG pathways and classification of metabolic pathways.

**Figure 8 ijms-26-04140-f008:**
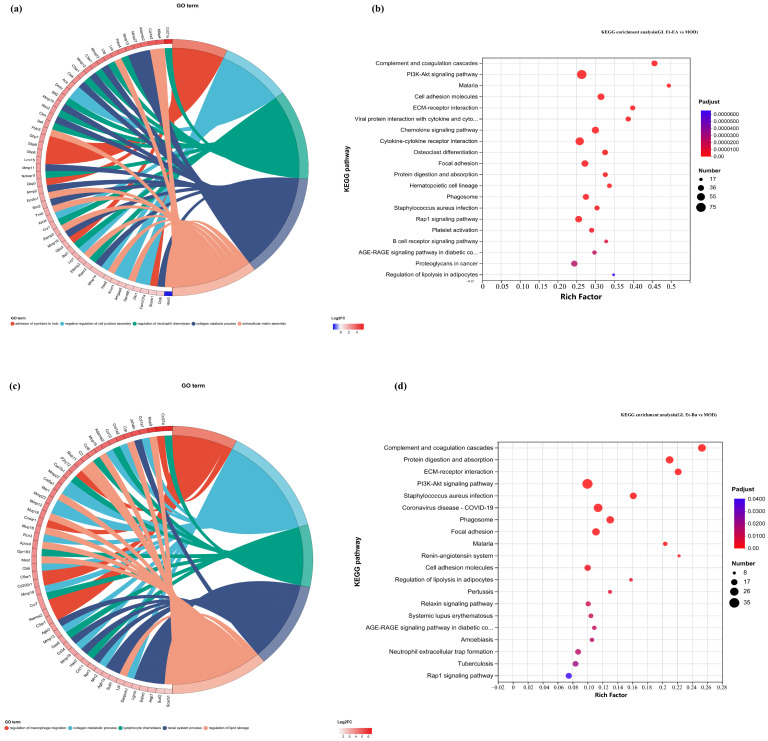
GO chord diagrams and KEGG enrichment diagrams of (**a**,**b**) G.L. ET-EA (E group), (**c**,**d**) G.L. ET-BU (F group), (**e**,**f**) G.L. WE-EA (G group), and (**g**,**h**) G.L. WE-BU (H group).

**Figure 9 ijms-26-04140-f009:**
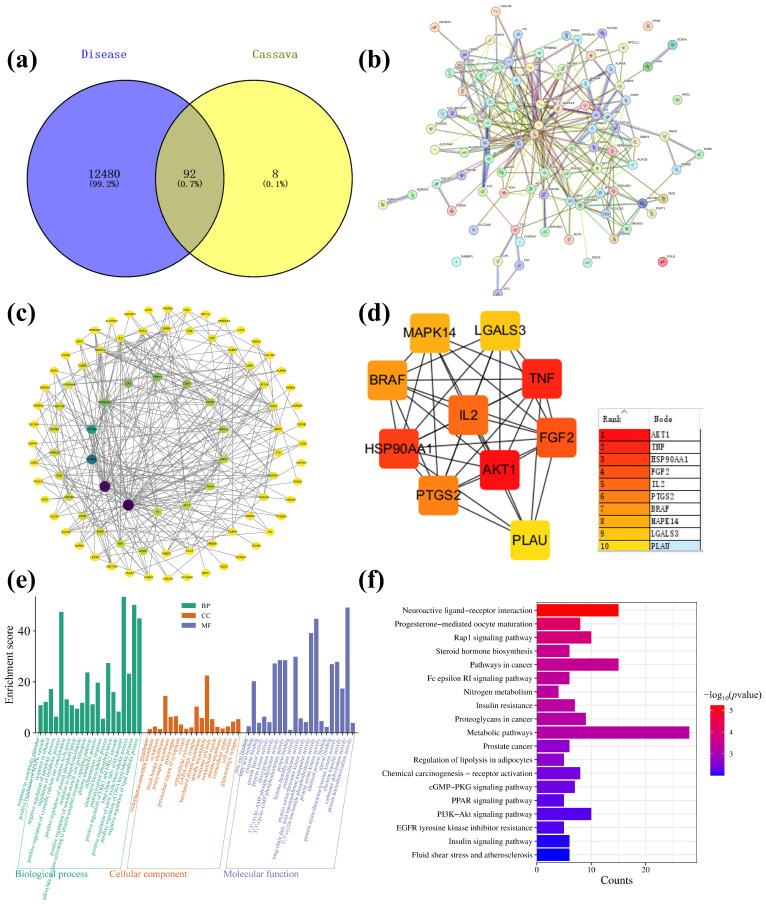
(**a**) Cassava leaf extracts, atopic dermatitis, and skin repair intersection gene Venn diagram; (**b**) PPI network diagram; (**c**) “Component-Target-Pathway” network diagram; (**d**) enrichment of top 10 core targets; (**e**) GO enrichment analysis diagram of cassava leaf extract in relation to skin inflammation; (**f**) KEGG enrichment analysis diagram of cassava leaf extract in relation to skin inflammation.

**Figure 10 ijms-26-04140-f010:**
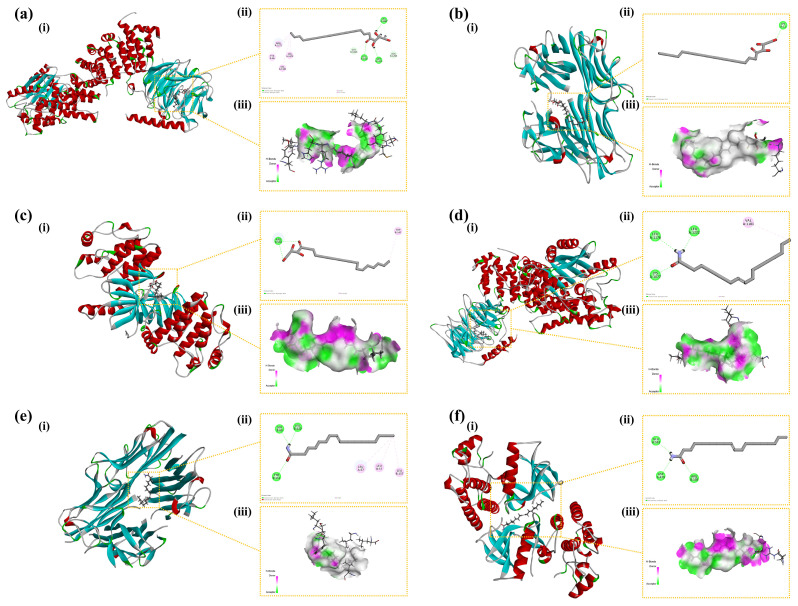
Binding modes of the compounds with the target proteins: (**a**) AKT1 with 1-stearoylglycerol; (**b**) TNF with 1-stearoylglycerol; (**c**) BRAF with 1-stearoylglycerol; (**d**) AKT1 with oleamide; (**e**) TNF with oleamide; and (**f**) BRAF with oleamide. Each panel is divided into three sections: (**i**) the 3D structure of the complex, (**ii**) the 2D binding mode of the complex, and (**iii**) the hydrogen bond donor–acceptor network of the complex. AKT1: Protein Kinase B alpha; TNF: Tumor Necrosis Factor; BRAF: B-Raf proto-oncogene, serine/threonine kinase.

**Table 1 ijms-26-04140-t001:** CDOCKER Energy of target proteins and compounds.

Name	CDOCKER Energy (kcal/mol)	
	1-Stearoylglycerol	Oleamide
AKT1	−48.4294	−29.8226
TNF	−33.5632	−16.8593
BRAF	−45.9103	−27.4083

**Table 2 ijms-26-04140-t002:** The names, extraction site, voucher specimen numbers, collection time, and storage location of the two plants.

Name	Extraction Site	Voucher Number	Collection Time	Storage Location
South China No. 9 leaves (G.L.)	Leaves	2022-S001	September 2022	School of Biomedical and Pharmaceutical Sciences, Guangdong University of Technology
Ziyehuangxin leaves (P.L.)	Leaves	2022-Z002	September 2022	

**Table 3 ijms-26-04140-t003:** Cassava species and the extraction methods used in this study.

Extraction Method/ Cassava Species	Ethanol Extraction Followed by Ethyl Acetate Extraction	Ethanol Extraction Followed by Butanol Extraction	Water Extraction Followed by Ethyl Acetate Extraction	Water Extraction Followed by Butanol Extraction
Ziyehuangxin leaves (P.L.)	A (ET-EA)	B (ET-BU)	C (WE-EA)	D (WE-BU)
South China No. 9 leaves (G.L.)	E (ET-EA)	F (ET-BU)	G (WE -EA)	H (WE-BU)

## Data Availability

Data will be made available on request.

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
