# Peer review of "A Comprehensive Analysis of Chemical Composition and Anti-Inflammatory Effects of Cassava Leaf Extracts in Two Varieties in Manihot esculenta Crantz"

_ijms, 2025, doi:10.3390/ijms26094140_

Round 1

Reviewer 1 Report

Comments and Suggestions for Authors

In general, the article presents a study on two variations of cassava (Manihot esculenta), from which eight samples were obtained according to the type of extraction of the compounds present in their tissues.

Although it is possible to observe extensive and significant work regarding biological assays, the description of the activities involving the determination of the chemical profile of the samples has important gaps that must be reviewed.

In the Abstract, some acronyms that are not so common to those who do not work with biological activity assays, such as HE and IHC, are not explained. They should be explained in the text.

Also in the Abstract, it is important to indicate whether the bioactive compounds cited are or may be responsible for the biological activities observed. After reading the manuscript, it is not clear why these compounds were chosen, specifically in this part of the work.

The description of the extraction methods needs to be reviewed. It is not clear whether the second extraction (what is the purpose of this extraction?) was done separately with ethyl acetate and butanol, or using a 1:1 mixture of ethyl acetate:butanol. This mixture is mentioned in the text, but Table 1 shows the results of the samples coded as if they had been prepared with the solvents separately.

The description of the analysis methodology by UPLC-Q-TOF/MS does not specify which solvents A and B are.

The final paragraph of the discussion of item 3.1 (p. 5, l.200 and 201) shows that the extraction treatment with ET-EA was more efficient than WEBU. What parameters is this conclusion based on? Qualitative? Quantitative? I suggest that this observation be based on the terms that the authors used to reach this conclusion.

The lack of detection, or at least citation of the annotation of cyanogenic glycosides in the manuscript is curious, given that these are compounds considered even as markers of the species and may even be related to the biological activities observed.

In general, the article does not provide a critical analysis of each fraction, correlating their respective chemical profiles with the results achieved in the anti-inflammatory assays. Obviously, a more accurate conclusion takes into account the role of each compound in the mechanism of action of the activity. However, it would be interesting to correlate the chemical composition of each sample with the observed activities.

The choice of substances to discuss the attribution of their possible role in the anti-inflammatory activity is precarious and I believe that it could be better discussed. Were the major compounds, according to the UPLC-Q-TOF/MS analyses, not noted? Does the literature provide any information about anti-inflammatory activities correlated with them?

I strongly suggest that the entire manuscript be reviewed and the suggestions be discussed so that the work can finally be considered for publication in the IJMS.

Comments on the Quality of English Language

English could be improved to make some points easier to understand.

Author Response

In general, the article presents a study on two variations of cassava (Manihot esculenta), from which eight samples were obtained according to the type of extraction of the compounds present in their tissues.

Although it is possible to observe extensive and significant work regarding biological assays, the description of the activities involving the determination of the chemical profile of the samples has important gaps that must be reviewed.

Response: Thank you for pointing out this issue. In this article, we primarily focused on the results of bioassays, while the specific correlation analysis between the chemical characteristics of the samples and their biological activities was relatively limited. This is mainly due to the chemical complexity of the samples and the limitations of the existing data, which prevented us from fully exploring the role of chemical components in the activity mechanisms. We plan to employ more advanced chemical analysis techniques (such as mass spectrometry, nuclear magnetic resonance, etc.) in subsequent studies to provide a more comprehensive characterization of the chemical composition of the samples, thereby addressing the current gaps in the description of chemical features. 

In the Abstract, some acronyms that are not so common to those who do not work with biological activity assays, such as HE and IHC, are not explained. They should be explained in the text.

Response: Thank you for pointing out this problem. We have explained the abbreviations in the abstract, as detailed in lines 21-28.

Also in the Abstract, it is important to indicate whether the bioactive compounds cited are or may be responsible for the biological activities observed. After reading the manuscript, it is not clear why these compounds were chosen, specifically in this part of the work.

Response: Thank you for your comment. We selected these compounds because they have a high proportion in the composition, and there are documented reports of their anti-inflammatory properties. In both the results and discussion, there are papers that can prove the anti-inflammatory effects of the relevant substances.

The description of the extraction methods needs to be reviewed. It is not clear whether the second extraction (what is the purpose of this extraction?) was done separately with ethyl acetate and butanol, or using a 1:1 mixture of ethyl acetate:butanol. This mixture is mentioned in the text, but Table 1 shows the results of the samples coded as if they had been prepared with the solvents separately.

Response: Thank you for your comment. We initially employed ethanol for extraction, followed by the addition of an equal volume of ethyl acetate to the ethanol extract for further extraction. We have already revised the wording, as detailed in lines 411-416.

The description of the analysis methodology by UPLC-Q-TOF/MS does not specify which solvents A and B are.

Response: Thank you for pointing out this problem. In the original expression, the default solvent written in the front is A (water (containing 0.1% formic acid)), but it is not clearly marked, and now it is clearly marked that A is water (containing 0.1% formic acid) and B is methanol, as detailed in lines 431-432.

The final paragraph of the discussion of item 3.1 (p. 5, l.200 and 201) shows that the extraction treatment with ET-EA was more efficient than WEBU. What parameters is this conclusion based on? Qualitative? Quantitative? I suggest that this observation be based on the terms that the authors used to reach this conclusion.

Response: Thank you for pointing out this problem. This conclusion was drawn from the subsequent activity assays. Upon reflection, we realized that this statement was not appropriate in this context, so we have already removed it.

The lack of detection, or at least citation of the annotation of cyanogenic glycosides in the manuscript is curious, given that these are compounds considered even as markers of the species and may even be related to the biological activities observed.

Response: Thank you for pointing out this problem. Since no cyanide was detected under our four extraction methods, this may be related to the extraction methods, but the specific reasons need to be further explored, and relevant contents have been added in the discussion section, as detailed in lines 312-317.

In general, the article does not provide a critical analysis of each fraction, correlating their respective chemical profiles with the results achieved in the anti-inflammatory assays. Obviously, a more accurate conclusion takes into account the role of each compound in the mechanism of action of the activity. However, it would be interesting to correlate the chemical composition of each sample with the observed activities.

Response: Thank you for pointing out this issue. It is clear that the main focus of this article is to investigate the potential anti-inflammatory effects and mechanisms of two cassava extracts. We believe that linking the chemical composition of each sample to the observed anti-inflammatory activity would be an interesting research direction, as it would help to deepen the understanding of the relationship between chemical components and biological activity. We will focus on this aspect in our future work. 

The choice of substances to discuss the attribution of their possible role in the anti-inflammatory activity is precarious and I believe that it could be better discussed. Were the major compounds, according to the UPLC-Q-TOF/MS analyses, not noted? Does the literature provide any information about anti-inflammatory activities correlated with them?

Response: Thank you for your comment. We have added papers to the results that demonstrate the anti-inflammatory effects of the compounds.

I strongly suggest that the entire manuscript be reviewed and the suggestions be discussed so that the work can finally be considered for publication in the IJMS.

Comments on the Quality of English Language

English could be improved to make some points easier to understand.

Response: Thank you for your comment. We have reviewed the entire manuscript and improved the English expression.

Reviewer 2 Report

Comments and Suggestions for Authors

This work (Title: A comprehensive analysis of chemical composition and anti-inflammatory effects of cassava leaf extracts in two varieties in Manihot esculenta Crantz, Manuscript Number: IJMS- 3486115) has analyzed components of South China No.9 (green leaves (G.L.)) and South China No.20 (purple leaves 19 (P.L.)) leaves using UPLC-Q-TOF/MS and found that cassava leaf extracts are rich in bioactive metabolites. Meanwhile, the anti-inflammatory efficacy of bioactive compounds has been assessed with animal models using HE staining, T.B. staining, and IHC staining. This result indicates that cassava leaf extracts seem to be promising natural anti-inflammatory agent. Thus, this reviewer is positive for publishing this manuscript if the authors have revised the manuscript accordingly to the following comments.

  • Define abbreviation when they were first appeared in the title, abstract, early in manuscript, such as UPLC-Q-TOF/MS, T.B. , and HE in Abstract.
  • “Less is known about the chemical composition and biological activity of cassava leaves.” The relation references need to discuss.
  • The chemical compositions of cassava can be shown and compared with the chemical compositions of cassava leaves.
  • The application of the chemical compositions of cassava using in food and medicine can be discussed.
  • The CAS number can be added into Tables S1 and S2.
  • In Figure 1, the section was labeled as A, B, and so on. However, A, B, C, and so on also had been applied in Figure 1A. Difference labels can be applied in Figure 1. This also can be found in other figures.
  • Figures 3 and 4 can be merged.
  • The word size in Figures 5, 6, 7, 8, and 9 can be increased to show clear.
  • The discussion for this work is weaken.
  • The limitation of this work needs to be considered.

Comments on the Quality of English Language

The English could be improved to more clearly express the research.

Author Response

Define abbreviation when they were first appeared in the title, abstract, early in manuscript, such as UPLC-Q-TOF/MS, T.B. , and HE in Abstract.

Response: Thank you for pointing out this problem. We have explained the abbreviations in the abstract, as detailed in lines 21-28.

“Less is known about the chemical composition and biological activity of cassava leaves.” The relation references need to discuss.

Response: Thank you for pointing out this problem. However, it is through our literature review. we found that the research on the bioactivity of cassava is mainly focused on the roots and tubers, while the studies on the leaves are relatively scarce. Therefore, we are unable to cite some papers to discuss them. But before that we have a paper that cited research on cassava roots.

The chemical compositions of cassava can be shown and compared with the chemical compositions of cassava leaves.

Response: Thank you for your comment. We have added a comparison of cassava and cassava leaves in the discussion section, as detailed in lines 54-66.

The application of the chemical compositions of cassava using in food and medicine can be discussed.

Response: Thank you for your comment. We have already discussed this part in the article, as detailed in lines 45-69.

The CAS number can be added into Tables S1 and S2.

Response: Thank you for your suggestion. We have the CAS number in Table S1 and Table S2.

In Figure 1, the section was labeled as A, B, and so on. However, A, B, C, and so on also had been applied in Figure 1A. Difference labels can be applied in Figure 1. This also can be found in other figures.

Response: Thank you for pointing out this problem. We also realized that this was not appropriate, so we changed the AB of the picture to ab.

Figures 3 and 4 can be merged.

Response: Thank you for pointing out this problem. We have merged Figure 3 and 4.

The word size in Figures 5, 6, 7, 8, and 9 can be increased to show clear.

Response: Thank you for pointing out this problem. We have improved the clarity of the picture.

The discussion for this work is weaken.

Response: Thank you for your comment. We have already added the discussion section, as detailed in lines 312-317 and 375-394.

The limitation of this work needs to be considered.

Response: Thank you for pointing out this problem. We have added the limitation in discussion, as detailed in lines 312-317 and 375-394.

Reviewer 3 Report

Comments and Suggestions for Authors

This manuscript by Cai, Jie et al investigates the chemical composition and anti-inflammatory effects of cassava leaf extracts from two South China varieties: No.9 (green leaves, G.L.) and No.20 (purple leaves, P.L.). Utilizing UPLC-Q-TOF/MS, researchers identified bioactive metabolites such as D-(+)-mannose, trigonelline, rutin, kaempferol-3-O-rutinoside, and oleamide. In mouse models, cassava leaf extracts demonstrated anti-inflammatory properties, evidenced by reduced epidermal thickness, fewer mast cells, and lower levels of inflammatory cytokines (IL-17 and TNF-a). The authors also performed transcriptomic analysis which suggested that these effects might be linked to the modulation of genes involved in mast cell activation, such as Cma1, Cpa3, and Fn1.

While the authors provided a plethora of data, especially with the animal model, there are some major drawbacks of this studies. There is a disconnection between the first part of the manuscript, UPLC-MS analysis of chemical composition of leaf extracts, and the later part of the biological activity study. The authors did not provide a good reason for studying samples differ in extraction methods. As the UPLC-MS analysis showed, most of the bioactive compounds exist in all the extracts from different extraction methods, differing only on the relative amount. Performing biological study with the whole extracts provided little information on molecular scale. It would be better to perform biological studies with, for example, different HPLC fractions.

There are several mistakes in method and data presentation, raising the concern on the overall data reliability. For example, in line 68-72, the author states that the extraction was performed with 1:1 mixture of EA and BU, but according to the later part of the manuscript, it should be either EA or BU, which gave separate samples. Also in Figure 7, sample G1-G3 were place in panel (A) where it is supposed to be sample D1-D3. Such errors in those critical part of the manuscript indicate the lack of rigorous in this scientific work.

In general, the authors seem to be more interested in just showing as much data as possible, rather than actually interpret the data. The anti-inflammatory effect is likely to be real, but the entire transcriptomics section did not provide further information other than listing of inflammation-related gene and pathways. The discussion on the RNAseq data is extremely confusing. For example, the upregulation of Cma1 gene (involved in mast cell inflammation response) compared to MOD is contradictory to the “anti-inflammatory” effects claimed for the leaf extracts. In addition, the entire GO and KEGG analysis section is unreadable due to the low resolution of the figures.

Also, in some of the biological effects evaluation experiments, the positive control group (DPH) show very little/insignificant effect (figure 1,2,4), raising the question whether these assays were properly chosen or performed.

The introduction section is obviously not sufficient and more on prior art of the field is required.

Comments on the Quality of English Language

Relatively easy to understand but still has space for improvement.

Author Response

While the authors provided a plethora of data, especially with the animal model, there are some major drawbacks of this studies. There is a disconnection between the first part of the manuscript, UPLC-MS analysis of chemical composition of leaf extracts, and the later part of the biological activity study. The authors did not provide a good reason for studying samples differ in extraction methods. As the UPLC-MS analysis showed, most of the bioactive compounds exist in all the extracts from different extraction methods, differing only on the relative amount. Performing biological study with the whole extracts provided little information on molecular scale. It would be better to perform biological studies with, for example, different HPLC fractions.

Response: Thank you for pointing out the issue. It is clear that the primary focus of this article is to investigate the biological activities of cassava extracts and to analyze the main components within these extracts. As for your suggestion regarding the use of different HPLC fractions for biological studies, we will consider this approach in our future work.

There are several mistakes in method and data presentation, raising the concern on the overall data reliability. For example, in line 68-72, the author states that the extraction was performed with 1:1 mixture of EA and BU, but according to the later part of the manuscript, it should be either EA or BU, which gave separate samples. Also in Figure 7, sample G1-G3 were place in panel (A) where it is supposed to be sample D1-D3. Such errors in those critical part of the manuscript indicate the lack of rigorous in this scientific work.

Response: Thank you for your comment. We have rewritten the experiment section, as detailed in lines 411-416. And we have corrected the error in Figure 6 (original figure 7).

In general, the authors seem to be more interested in just showing as much data as possible, rather than actually interpret the data. The anti-inflammatory effect is likely to be real, but the entire transcriptomics section did not provide further information other than listing of inflammation-related gene and pathways. The discussion on the RNAseq data is extremely confusing. For example, the upregulation of Cma1 gene (involved in mast cell inflammation response) compared to MOD is contradictory to the “anti-inflammatory” effects claimed for the leaf extracts. In addition, the entire GO and KEGG analysis section is unreadable due to the low resolution of the figures.

Response: Thank you for pointing out this problem. We have supplemented the relevant content, which can be found in lines 213-224 and 241-249 of the manuscript. Additionally, we have enhanced the image resolution for the GO and KEGG analysis sections.

Also, in some of the biological effects evaluation experiments, the positive control group (DPH) show very little/insignificant effect (figure 1,2,4), raising the question whether these assays were properly chosen or performed.

Response: Thank you for bringing this issue to our attention. Regarding the positive control group showing a very small or insignificant effect, we believe that this could be the result of a combination of individual differences among the mice and an insufficient sample size.

The introduction section is obviously not sufficient and more on prior art of the field is required.

Response: Thank you for your comment. We've perfected the introduction, as detailed in lines 45-69.

Reviewer 4 Report

Comments and Suggestions for Authors

The manuscript “A comprehensive analysis of chemical composition and anti-inflammatory effects of cassava leaf extracts in two varieties in Manihot esculenta Crantz” by Cai and coworkers describes the phytochemical and biological content analysis of two varieties of cassava. Phytochemical analysis consisted of UPLC-Q-TOF/MS, whereas biological evaluation was based on anti-inflammatory capacity assays. In addition, this study included the evaluation of the anti-inflammatory activity of compounds in animal models. The data provided in the submitted version is also based on transcriptomic analyses which were performed to investigate genes (Cma1, Cpa3, and Fn1) involved in the anti-inflammatory activity of cassava leaves extracts. The manuscript is interesting, but the research novelty is low since cassava extracts have been widely reported over the last decades because of their valuable therapeutic compounds, which have been already validated through in vitro, in vivo, and in silico analysis. The current version requires additional efforts to be considered for publication, especially in data and figure presentation together with information required to provide more detailed explanation about the need of what is being investigated.

The introduction requires to present a background about inflammatory disorders, considering their epidemiology and the limitations of current treatment regimens. In addition, the authors are commended to include information about traditional medicine and the advantages of considering natural products for drug development. The aim of this study should be revised by mentioning the main assays, and key results.

In section 2.2, the authors are advised to include the voucher identification of the two investigated species.

The authors are encouraged to represent the modeling and drug delivery assays with animals. The following article can be consulted to improve the quality of such section: 10.3390/antiox13060664.

In L. 115, the activity of histamine can be compared with the cassava leaf extracts? This is important to consider since the former was applied intradermal whereas the latter was applied topical. The differences between the administration can be associated with differences in terms of efficacy and systemic absorption.

Figures 5, 6 and 7 need to be revised since in their current version, the text can be barely seen. The same for figure 8 and 9. The discussion section can be improved by separating paragraphs. In addition, it can be benefited if the authors discuss how the structure of the identified compounds influence anti-inflammatory responses.

Revise the references section since various of them are outdated (e.g., 1996 and 1973).

Comments on the Quality of English Language

None.

Author Response

The manuscript “A comprehensive analysis of chemical composition and anti-inflammatory effects of cassava leaf extracts in two varieties in Manihot esculenta Crantz” by Cai and coworkers describes the phytochemical and biological content analysis of two varieties of cassava. Phytochemical analysis consisted of UPLC-Q-TOF/MS, whereas biological evaluation was based on anti-inflammatory capacity assays. In addition, this study included the evaluation of the anti-inflammatory activity of compounds in animal models. The data provided in the submitted version is also based on transcriptomic analyses which were performed to investigate genes (Cma1, Cpa3, and Fn1) involved in the anti-inflammatory activity of cassava leaves extracts. The manuscript is interesting, but the research novelty is low since cassava extracts have been widely reported over the last decades because of their valuable therapeutic compounds, which have been already validated through in vitro, in vivo, and in silico analysis. The current version requires additional efforts to be considered for publication, especially in data and figure presentation together with information required to provide more detailed explanation about the need of what is being investigated.

Response: Thank you for your comment. Cassava leaves as feed, its added value itself is low, there is no need to deliberately seek innovation. However, from the perspective of sustainable development, if the active ingredients can be extracted, not only can avoid the idle and waste of resources, but also significantly enhance its economic value. It is also a highly innovative and forward-looking initiative. We have refined the way the data and charts are presented and provided detailed explanations and required information about what is being investigated.

The introduction requires to present a background about inflammatory disorders, considering their epidemiology and the limitations of current treatment regimens. In addition, the authors are commended to include information about traditional medicine and the advantages of considering natural products for drug development. The aim of this study should be revised by mentioning the main assays, and key results.

Response: Thank you for your comment. We have provided background on inflammatory diseases in the introduction, complementing the limitations of current treatment regimens for dermatitis, as detailed in lines 45-69.

In section 2.2, the authors are advised to include the voucher identification of the two investigated species.

Response: Thank you for pointing out this problem. We have added the voucher identification of the two investigated species., as detailed in Table 1.

The authors are encouraged to represent the modeling and drug delivery assays with animals. The following article can be consulted to improve the quality of such section: 10.3390/antiox13060664.

Response: Thank you for pointing out this problem. We have already discussed this issue and cited this paper in the discussion section, as detailed in lines 382-385.

In L. 115, the activity of histamine can be compared with the cassava leaf extracts? This is important to consider since the former was applied intradermal whereas the latter was applied topical. The differences between the administration can be associated with differences in terms of efficacy and systemic absorption.

Response: Thank you for your comment. This type of experimental design is common in natural product studies, because many disease models rely on subcutaneous injection modeling, and natural products are mainly applied topically in cosmetics. To evaluate the experimental results more fully, we have added to the revised manuscript a discussion of potential limitations in comparing different routes of administration and highlight the need for future studies to further validate their efficacy and absorption properties, as detailed in lines 382-385.

Figures 5, 6 and 7 need to be revised since in their current version, the text can be barely seen. The same for figure 8 and 9. The discussion section can be improved by separating paragraphs. In addition, it can be benefited if the authors discuss how the structure of the identified compounds influence anti-inflammatory responses.

Response: Thank you for your comment. We have enhanced the clarity of the images and addressed the issues you raised.

Revise the references section since various of them are outdated (e.g., 1996 and 1973).

Response: Thank you for pointing out this problem. We have updated the references.

Reviewer 5 Report

Comments and Suggestions for Authors

The authors  tried to analyze the chemical composition and bioactive effects of cassava leaves using  UPLC-Q-TOF/MS . The experiments is well designed and the results are sufficient to support the hypothesis. The manuscript can be accepted before some minor issues being addressed.

1\ Add some antioxidant activity testing experiments,as antioxidant activity is often associated with anti-inflammatory activity.

2\Can you identify several key molecules that produce anti-inflammatory activity?

3\ It is highly recommended to provide physical photos of cassava leaf and its different extraction products in the article.

4\ The introduction section should provide a more detailed research background and highlight the innovation of this work.

5\Under testing at capillary temperature exceeding 300 degrees Celsius, will this affect the identification of thermally unstable compounds in the extraction solution?

6\ Language should be further optimized.

7\ The text in Figure 8 is difficult to read clearly, please optimize it.

Author Response

1\ Add some antioxidant activity testing experiments,as antioxidant activity is often associated with anti-inflammatory activity.

Response: Thank you for your comment. Antioxidant activity does usually correlate with anti-inflammatory activity, but we ran out of samples and the cassava was harvested, the raw material is not available at the moment, but we are adding that to the discussion, as detailed in lines 388-394.

2\Can you identify several key molecules that produce anti-inflammatory activity?

Response: Thank you for bringing this to our attention. We believe that the key molecules with definitive anti-inflammatory activity include: C1qtnf3, Cma1, Cpz, Fbn1, Fn1, Mcpt4, Serpina3c, Serpina3n, Lep, Adipoq, Cfd, Cfh, Ccl21a, Hp, and Lbp. These molecules exert their anti-inflammatory effects by modulating inflammatory signaling pathways, inhibiting the production of inflammatory mediators, or regulating immune cell functions. We have supplemented the relevant content, which can be found in lines 213-224 and 241-249 of the manuscript.

3\ It is highly recommended to provide physical photos of cassava leaf and its different extraction products in the article.

Response: Thank you for your comment. We have rewritten the experimental part because there may be some misunderstanding about the sample due to the expression problems in the experimental part, as detailed in lines 411-416. Since our samples have been used up and cassava has been harvested, raw materials are not available at present, so we are sorry for that we cannot provide photos of extractions.

4\ The introduction section should provide a more detailed research background and highlight the innovation of this work.

Response: Thank you for pointing out this problem. We've perfected the introduction. Cassava leaves as feed, its added value itself is low, there is no need to deliberately seek innovation. However, from the perspective of sustainable development, if the active ingredients can be extracted, not only can avoid the idle and waste of resources, but also significantly enhance its economic value. It is also a highly innovative and forward-looking initiative.

5\Under testing at capillary temperature exceeding 300 degrees Celsius,will this affect the identification of thermally unstable compounds in the extraction solution?

Response: Thank you for pointing out this issue. In tests where the capillary temperature exceeds 300°C, it is indeed possible that the identification of thermally unstable compounds in the extraction solution could be affected, potentially leading to their degradation, volatilization, or transformation, thereby reducing the accuracy and sensitivity of detection. In light of this, we are considering optimizing the temperature conditions in future studies to facilitate the identification of a greater number of thermally unstable compounds. 

6\ Language should be further optimized.

Response: Thank you for your comment. We have improved the language.

7\ The text in Figure 8 is difficult to read clearly, please optimize it.

Response: Thank you for pointing out this problem. We have improved the resolution of the picture.

Round 2

Reviewer 1 Report

Comments and Suggestions for Authors

All the suggestions were considered by the authors and either explained or corrected in the manuscript. Thus, I strongly suggest its publication.

Author Response

Dear Reviwer,

We thank you very much for giving us the opportunity to revise our manuscript. At the same time, we very much appreciate your positive and constructive comments and suggestions on our manuscript.  During this round, we have carefully considered the comments and have made corrections that we hope will be approved. Revised portions are marked in red on paper. Please chenk the attached file. We hope you everything goes well.

Best Wishes,

Feifei An

Reviewer 3 Report

Comments and Suggestions for Authors

Reviewer's response to authors’ response:

“Response: Thank you for pointing out the issue. It is clear that the primary focus of this article is to investigate the biological activities of cassava extracts and to analyze the main components within these extracts. As for your suggestion regarding the use of different HPLC fractions for biological studies, we will consider this approach in our future work.”

This reviewer understood that. However, there is a disconnection between the component analysis and the biological effects among different exacts, where the authors did not try to provide analysis on what molecular difference in different extracts resulted the different biological effects. This makes the manuscript less relevant to the “molecular science” focus of the journal and really limited the significance of this study.

“Response: Thank you for your comment. We have rewritten the experiment section, as detailed in lines 411-416. And we have corrected the error in Figure 6 (original figure 7).”

Thanks for revising that. In the updated method, the author wrote “The mixture was heated at 50°C for 3 hours. The concentrate was then divided into two portions and subjected to extraction again with an equal volume of ethyl acetate (EA) or n-butanol (BU)”. This reviewer is genuinely confused on how that worked. The “concentrate” was in ethanol right? Ethanol is miscible with EA or butanol. How could the second extraction work?

“Response: Thank you for pointing out this problem. We have supplemented the relevant content, which can be found in lines 213-224 and 241-249 of the manuscript. Additionally, we have enhanced the image resolution for the GO and KEGG analysis sections.”

The authors still haven’t answered the original question on the gene effects. For example, based on the volcano plot, the Cma1 gene is upregulated, correct? And Cma1 is involved in the generation of inflammatory mediators as the author stated. So upregulation of Cma1 should be pre-inflammatory, not anti-inflammatory, right? This is contradictory to the effect the authors tried to prove.

“Response: Thank you for bringing this issue to our attention. Regarding the positive control group showing a very small or insignificant effect, we believe that this could be the result of a combination of individual differences among the mice and an insufficient sample size.”

So the author admitted that the test group may also suffer from “individual differences among the mice and an insufficient sample size”? Then how could one believe the results are valid? Unless the authors believe that the Cassava extract samples are much more potent than the positive control, which is very unlikely.

Author Response

Dear Reviwer,

We thank you very much for giving us the opportunity to revise our manuscript. At the same time, we very much appreciate your positive and constructive comments and suggestions on our manuscript.  During this round, we have carefully considered the comments and have made corrections that we hope will be approved. Revised portions are marked in red on paper.  Please find below our answers  and detailed information on the changes made.

Comments 1: This reviewer understood that. However, there is a disconnection between the component analysis and the biological effects among different exacts, where the authors did not try to provide analysis on what molecular difference in different extracts resulted the different biological effects. This makes the manuscript less relevant to the “molecular science” focus of the journal and really limited the significance of this study.

“Response: Thank you for pointing out the issue. We acknowledge that a more detailed molecular analysis could strengthen the relevance of our study to the journal's focus on molecular science. To address this concern, we have supplemented additional experiments to analyze the relationship between the molecular characteristics and biological effects of different extracts using network pharmacology and molecular docking techniques. These analyses have revealed specific molecular differences and their correlations with the observed biological effects. We have included these findings in the revised manuscript, along with a discussion on how these molecular differences may contribute to the varying biological activities. We hope that these additional analyses and discussions will enhance the molecular science aspect of our study and underscore its significance. Thank you for your valuable feedback, which has greatly improved the quality of our manuscript.

Comments 2: Thanks for revising that. In the updated method, the author wrote “The mixture was heated at 50°C for 3 hours. The concentrate was then divided into two portions and subjected to extraction again with an equal volume of ethyl acetate (EA) or n-butanol (BU)”. This reviewer is genuinely confused on how that worked. The “concentrate” was in ethanol right? Ethanol is miscible with EA or butanol. How could the second extraction work?

“Response: Thank you for pointing out the issue. We regret any confusion caused by our initial description. We have revised the methodology accordingly, as detailed in lines 533-543 of the manuscript.

Comments 3: The authors still haven’t answered the original question on the gene effects. For example, based on the volcano plot, the Cma1 gene is upregulated, correct? And Cma1 is involved in the generation of inflammatory mediators as the author stated. So upregulation of Cma1 should be pre-inflammatory, not anti-inflammatory, right? This is contradictory to the effect the authors tried to prove.

“Response: Thank you for your thoughtful question regarding the role of Cma1 in our study. You are correct that Cma1 is generally considered a pro-inflammatory gene due to its involvement in generating inflammatory mediators such as angiotensin II and its role in mast cell degranulation. However, the relationship between gene expression and functional outcomes is complex and context-dependent. Several factors may explain the apparent contradiction: 1, While Cma1 is typically pro-inflammatory, its role may vary depending on the cellular environment, experimental conditions, and interactions with other factors, potentially leading to an overall anti-inflammatory effect in our study. 2, The broader gene network, including the upregulation of anti-inflammatory genes (e.g., IL-10 or TGF-β), might counteract Cma1's pro-inflammatory effects. 3, Cma1's role may differ between acute and chronic inflammation, with its upregulation in our experimental context potentially contributing to tissue repair or inflammation resolution. 4, The volcano plot reflects gene expression changes but does not directly indicate protein activity or functional outcomes, as post-transcriptional or post-translational regulation might modulate Cma1's effects. We acknowledge the importance of this point and agree that further mechanistic studies are needed to fully elucidate Cma1's role in the observed anti-inflammatory effect. We appreciate your attention to this detail and will address it in future research. The relevant content has been added to lines 241-247 in the manuscript.

Comments 4: So the author admitted that the test group may also suffer from “individual differences among the mice and an insufficient sample size”? Then how could one believe the results are valid? Unless the authors believe that the Cassava extract samples are much more potent than the positive control, which is very unlikely.

“Response: Thank you for pointing out the issue. We appreciate your attention to the potential issues of individual mouse variability and insufficient sample size in the experiments. Although these are common challenges in biological research, we ensured the reliability and validity of the results through rational experimental design, random grouping, rigorous statistical analysis, and the inclusion of control groups. The cassava extract demonstrated significant biological activity, though its effects were not intended to surpass those of the positive control but rather to evaluate its potential as an alternative. We have revised the manuscript to more clearly discuss these limitations and their implications, further enhancing the transparency and rigor of the study. The relevant content can be found in lines 505-513 of the revised manuscript.

We have carefully studied your comments and have made a revision that is marked in red in the paper. We have tried our best to revise our manuscript according to the comments. Attached please find the revised version, which we would like to submit for your kind consideration. We would like to express our great appreciation to you and your comments on our paper. Looking forward to hearing from you.

Thank you and best regards.

Yours sincerely,

Fei fei An

Reviewer 4 Report

Comments and Suggestions for Authors

The authors have addressed all my comments providing reasonable information and highlighting the novelty of this research in the revised version of the manuscript.

Author Response

(The authors gave the same response as above.)

Round 3

Reviewer 3 Report

Comments and Suggestions for Authors

Thank the authors for improving the manuscript.